



# Design and description of the MUSICA IASI full retrieval product

Matthias Schneider[1], Benjamin Ertl[1,2], Christopher Diekmann[1], Farahnaz Khosrawi[1], Andreas Weber[2,*], Frank Hase[1], Michael Höpfner[1], Omaira E. García[3], Eliezer Sepúlveda[3], and Douglas Kinnison[4]

[1]Institute of Meteorology and Climate Research (IMK-ASF), Karlsruhe Institute of Technology, Karlsruhe, Germany
[2]Steinbuch Centre for Computing (SCC), Karlsruhe Institute of Technology, Karlsruhe, Germany
[3]Izaña Atmospheric Research Center, Agencia Estatal de Meteorología (AEMET), Santa Cruz de Tenerife, Spain
[4]Atmospheric Chemistry Observations & Modeling Laboratory, National Center for Atmospheric Research, Boulder, Colorado, USA
[*]now at: Research & Development, Dynatrace Austria GmbH, Linz, Austria

**Correspondence:** M. Schneider
(matthias.schneider@kit.edu)

**Abstract.** IASI (Infrared Atmospheric Sounding Interferometer) is the core instrument of the currently three Metop (Meteorological operational) satellites of EUMETSAT (European Organization for the Exploitation of Meteorological Satellites). The MUSICA IASI processing has been developed in the framework of the European Research Council project MUSICA (MUlti-platform remote Sensing of Isotopologues for investigating the Cycle of Atmospheric water). The processor performs

an optimal estimation of the vertical distributions of water vapour ($H_2O$), the ratio between two water vapour isotopologues (the $HDO/H_2O$ ratio), nitrous oxide ($N_2O$), methane ($CH_4$), and nitric acid ($HNO_3$), and works with IASI radiances measured under cloud-free conditions in the spectral window between 1190 and $1400\,cm^{-1}$. The retrieval of the trace gas profiles is performed on a logarithmic scale, which allows the constraint and the analytic treatment of $\ln[HDO] - \ln[H_2O]$ as proxy for the $HDO/H_2O$ ratio. Currently, the MUSICA IASI processing has been applied to all IASI measurements available between

October 2014 and April 2020, so more than 1.4 billion individual retrievals have been performed.

Here we describe the MUSICA IASI full retrieval product data set. The data set is made available in form of netcdf data files that are compliant with version 1.7 of the CF (Climate and Forecast) metadata convention. For each orbit an individual standard output data file is provided. These files contain for each individual retrieval information on the a priori usage and constraint, the retrieved atmospheric trace gas and temperature profiles, profiles of the leading error components, information

on vertical representativeness in form of the averaging kernels as well as averaging kernel metrics, which are more handy than the full kernels. We discuss data filtering options and give examples of the high horizontal and continuous temporal coverage of the MUSICA IASI data products. The standard output data files provide comprehensive information for each individual retrieval resulting in a rather large data volume (about 25 TB for the more than five years of data with global daily coverage). This at a first glance apparent drawback of large data files and data volume is counterbalanced by multiple possibilities of data

reusability, which are briefly discussed.

In an extended output data file the same variables as in the standard output data files are provided in addition to Jacobians for many different uncertainty sources and Gain matrices (due to this additional variables it is called the extended output). It is limited to 74 observations over a polar, mid-latitudinal and tropical site. We use this additional Jacobian and Gain data



for assessing the typical impact of different uncertainty sources – like surface emissivity or spectroscopic parameters – and
different cloud types on the retrieval results.

We offer two data packages with DOI for free download via the repository RADAR4KIT. The first data package has a data
volume of about 17.5 GB and is linked to https://doi.org/10.35097/408 (Schneider et al., 2021b). It contains example standard
output data files for all MUSICA IASI retrievals made for a single day (more than 0.6 million). Furthermore, it includes a
ReadMe.pdf file with a description of how to access the total data set (the 25 TB) or parts of it. This data package is for users
interested in the typical global daily data coverage and in information about how to download the large data volumes of global
daily data for longer periods. The second data package is linked to https://doi.org/10.35097/412 (Schneider et al., 2021a) and
contains the extended output data file. Because it provides data for only 74 example retrievals, its data volume is only 73 MB
and it is thus recommended to users for having a quick look on the data.

# 1   Introduction

The IASI (Infrared Atmospheric Sounding Interferometer, a thermal nadir sensor, Blumstein et al., 2004) instrument aboard the
Metop (Meteorological Operational) satellites presents possibilities for measuring a large variety of different atmospheric trace
gases (e.g. Clerbaux et al., 2009) with a daily global coverage. Because each Metop is an operational EUMETSAT (European
Organization for the Exploitation of Meteorological Satellites) satellite, IASI measurements offer an excellent global daily
coverage and a sustained long-term perspective (measurements of IASI and IASI successor instruments are guaranteed between
2006 and the 2040s). This provides unique opportunities for consistent long-term observations and climate research.

In addition to humidity and temperature profiles (which are the meteorological core products, August et al., 2012) IASI
can detect, for instance, atmospheric ozone ($O_3$, e.g. Keim et al., 2009; Boynard et al., 2018), carbon monoxide (CO, e.g.
De Wachter et al., 2012), nitric acid ($HNO_3$, Ronsmans et al., 2016), nitrous oxide and methane ($N_2O$ and $CH_4$, De Wachter
et al., 2017; Siddans et al., 2017; García et al., 2018), the ratio between different water vapour isotopologues (Schneider and
Hase, 2011; Lacour et al., 2012) and different volatile organic compounds (Franco et al., 2018).

These diverse opportunities of IASI together with the good horizontal and daily coverage result in a large amount of IASI
products generated in the context of often computationally expensive retrievals. In oder to ensure ultimate benefit from these
efforts the generated data should be FAIR (e.g. Wilkinson et al., 2016): findable, accessible, interoperable, and reusable.

During the European Research Council project MUSICA (MUlti-platform remote Sensing of Isotopologues for investigating
the Cycle of Atmospheric water, from 2011 to 2016) we developed at the Karlsruhe Institute of Technology a processor for
the analysis of the thermal nadir spectra of IASI. Here we present the MUSICA IASI trace gas processing output, which
encompasses vertical profiles of $H_2O$, $\delta D$ ($\delta D = 1000 \left( \frac{HDO/H2O}{VSMOW} - 1 \right)$ with Standard Mean Ocean Water, $VSMOW = 3.1152 \times$





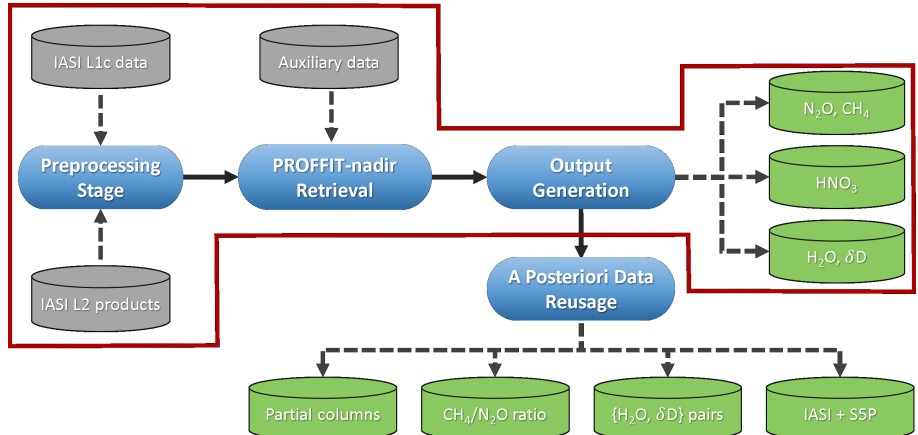

**Figure 1.** Outline of the MUSICA IASI processing chain. This paper focuses on the processing steps as indicated by the red frame. The green symbols indicate different products. The supply of detailed information on retrieval settings and product characteristics offers many different possibilities for a posteriori processing and date resusage (indicated on the bottom of the schematics and discussed in Sect. 8).

$10^{-4}$), $N_2O$, $CH_4$, and $HNO_3$. In addition to the retrieved trace gas profiles, the processing output consists of a comprehensive
set of variables describing the retrieval settings and product characteristics for each individual retrieval.

Figure 1 shows a schematic of the MUSICA IASI processing chain and data reusage possibilities. In this work we focus on the main processing chain, which is indicated by red frame in Fig. 1. In a preprocessing step EUMETSAT IASI spectra (L1c) and EUMETSAT IASI retrieval products (L2) are merged and observations made under cloudy conditions are filtered out. The EUMETSAT data and data from other sources (e.g. model data for the generation of the a priori information, emissivity
and topography data bases, spectroscopic parameters) serve then as input for the retrieval code PROFFIT-nadir. In the output generation stage the PROFFIT-nadir output is converted into netcdf data files following a well-known metadata standard. The data are easily findable via Digital Object Identifyers (DOIs) and are freely available for download at http://www.imk-asf.kit. edu/english/musica-data.php.

The integrated supply of comprehensive information on retrieval input and retrieval settings (measured spectra, used a
priori states, and constraints) and the retrieval output and characteristics (retrieved state vectors, averaging kernels, and error covariances) makes the data processing fully reproducible and strongly facilitates data interoperability and data reusage. Some examples are indicated on the bottom of the schematics of Fig. 1.

The paper is organised as follows: Section 2 briefly presents the satellite experiment on which the retrieval product relies. Section 3 describes the structure of the data files, the data volume, and the nomenclature of the data variables. In Sect. 4
we discuss the details of the MUSICA IASI retrieval setup. There we describe the cloud filtering and the comprehensive information that is provided about the a priori state vectors and the generation of the applied constraints. This information is essential for being able to perform an a posteriori processing according to Diekmann et al. (2021), or to optimally combine the data with other remote sensing data products (e.g. Schneider et al., 2021c). Section 4 can be skipped by readers that do not





plan such complex data reuse. In Section 5 the data variables and the variables describing the quality of the data are explained.
This is of general importance for correctly using the data (understanding uncertainties, representativeness, application in the context of model comparisons and data assimilation systems, application for inter-comparison studies, etc.). In Sect. 6 the options for filtering data according to their quality and characteristics are discussed. This enables the user to develop their own tailored data filtering. Section 7 visualises the data volume in form of two examples. A first example shows the continuous data availability over several years and a second example the good global daily data coverage. Section 8 discusses the potential of the data set in regard to data interoperability and data reuse, which is achieved by providing the retrieved state vectors together with comprehensive information on the a priori state vectors, the constraint matrices, the averaging kernels matrices, and the error covariance matrices. A summary and an outlook are provided in Sect. 9. For readers that are no experts in the field of remote sensing retrievals, Appendix A provides a short compilation with the theoretical basics and the most important equations on which we refer throughout this paper. Appendix B reveals that for the MUSICA IASI retrieval product we can assume moderate non-linearity (according to Chapter 5 of Rodgers, 2000), which is important for many data reuse options. Appendix C explains how the data can be used in form of a total or partial column product.

## 2 The IASI instruments on Metop satellites

IASI is a Fourier-transform spectrometer and measures in the infrared part of the electromagnetic spectrum between $645\,\mathrm{cm}^{-1}$ and $2760\,\mathrm{cm}^{-1}$ ($15.5\,\mu\mathrm{m}$ and $3.63\,\mu\mathrm{m}$). After apodisation (L1c spectra) the spectral resolution is $0.5\,\mathrm{cm}^{-1}$ (Full Width Half Maximum, FWHM). The main purpose of IASI is the support of Numerical Weather prediction. However, due to its high signal to noise ratio and the high spectral reolution, the IASI measurements offer very interesting possibilities for atmospheric trace gas observations (e.g. Clerbaux et al., 2009).

The IASI instruments are carried by the Metop satellites, which are Europe's first polar-orbiting satellites dedicated to operational meteorology. The Metop program has been planned as a series of three satellites to be launched sequentially over an observational period of 14 years. Metop-A was launched on 19 October 2006, Metop-B on 17 September 2012, and Metop-C on 7 November 2018. IASI is the main payload instrument and operates in the nadir viewing geometry with horizontal resolution of 12 km (pixel diameter at nadir viewing geometry) over a swath width of about 2200 km. With 14 orbits in a sun-synchronous mid-morning orbit (9:30 local solar time, LT, descending node), each IASI on a Metop satellite provides observations twice a day at middle and low latitudes (at about 9:30 and 21:30 LT) and several times a day at high latitudes. Until the beginning of 2020 the Metop-A, -B, and -C overflight times took place within about 45 minutes. Table 1 gives an overview on the major specifications of the Metop/IASI mission.

The number of individual observations made by the three currently orbiting IASI instruments is tremendous. During a single orbit 91800 observations are made. In 24 h the three satellites conclude in total about 42 orbits, which means more than 3.85 million individual IASI spectra per day and more than 1.4 billion per year.

IASI-like observations are guaranteed for several decades. First observations are made in 2006 and in context of the Metop Second Generation (Metop-SG) satellite programme IASI Next Generation instruments will perform measurements until the



**Table 1.** Overview on specifications of IASI on Metop.

| Type | Specification |
|---|---|
| Launch dates | Metop-A/IASI-A: October 19, 2006 |
|  | Metop-B/IASI-B: September 17, 2012 |
|  | Metop-C/IASI-C: November 7, 2018 |
| Altitude | 817 km |
| Orbit type | polar, sun-synchronous |
| Local overpass time | descending orbit: about 09:30 local time |
|  | ascending orbit: about 21:30 local time |
| IASI sensor | Fourier transform spectrometer |
| Spectral coverage | 645 to 2760 $cm^{-1}$ |
| Spectral resolution | 0.5 $cm^{-1}$ |
| Horizontal resolution | 12 km diameter of ground pixel at nadir |
| Full swath width | 2200 km |
| Orbit rate | 14 orbits per day and satellite |
| Global Earth coverage | 2 per day and satellite |

2040s. In this context the IASI programme offers unique possibilities for studying the long-term evolution of the atmospheric composition.

## 3 MUSICA IASI data format

In this section we discuss the format of the MUSICA IASI full product data files and the nomenclature of the data variables.

### 3.1 Data files

The MUSICA IASI full product data are provided as netcdf files compliant with version 1.7 of the CF (Climate and Forecast) metadata convention (cfconvention.org). The data files contain all information needed for reproducing the retrievals and for optimally reusing the data. Because the MUSICA IASI retrieval builds upon the EUMETSAT L2 cloud filter and uses the
115 EUMETSAT L2 atmospheric temperature as the a priori atmospheric temperature, the output files contain some EUMETSAT retrieval data as well as the MUSICA retrieval data. In addition, they contain the EUMETSAT L1C spectral radiances (and the simulated radiances) as well as auxiliary data needed for the retrieval (like surface emissivity from other sources, Masuda et al., 1988; Seemann et al., 2008; Baldridge et al., 2009).

We provide standard output files comprising all processed IASI observations and one extended output file with detailed
calculations of Jacobians and Gain matrices for a few selected observations.



The standard output is provided in the files `IASI[S]_MUSICA_[V]_L2_AllTargetProducts_[D]_[O].nc` and in one files per orbit and instrument. The symbols within the edged parenthesis indicate placeholders: `[S]` for the sensor (A, B, or C, for IASI instruments on the satellites Metop-A, -B, or -C, respectively), `[V]` for the used MUSICA IASI retrieval processor version, `[D]` for the starting date and time of the observation (format YYYYMMDDhhmmss), and `[O]` for the number of the orbit.

On our data base these files are provided in daily tar-files, with all orbits of all IASI instruments archived into a single tar-file, with the name `IASI[multipleS]_MUSICA_[V]_L2_AllTargetProducts_[DAY].tar`. The placeholders are: `[multipleS]` for the considered sensors, e.g. `AB` if IASI sensors on Metop-A and –B are considered, `[V]` for the used MUSICA IASI retrieval processor version, and `[DAY]` is the date of observations (Universal Time, format YYYYMMDD). The typical size of a tar-file with the orbit wise netcdf files of a single day is 15 GB. This number is for the typically 28 orbits per day of two satellite. The standard output data files are linked to a DOI (Schneider et al., 2021b).

The extended file represents 74 observations over polar, mid-latitudinal and tropical GRUAN stations (GRUAN stands for Global Climate Observing System Reference Upper Air Network, www.gruan.org). More details on the time periods and locations represented by these retrievals are given in Borger et al. (2018). The file provides the same output as the standard files and in addition detailed information on Jacobians and gain matrices. The name of this extendend output file is `IASIAB_MUSICA_030201_L2_AllTargetProductsExtended_examples.nc`, its size is 70 MB, and it is linked to an extra DOI (Schneider et al., 2021a).

Here we report on the MUSICA IASI processing version 3.2.1 (applied for IASI observations until the end of June 2019). For observations from July 2019 onward processing version 3.3.0 is applied. Both versions use the same retrieval setting and the output files contain the same variables. The difference between the versions is that for version 3.2.1 some minor correction have been made after the retrieval process. These corrections addressed some very minor inconsistencies in the vertical gridding, the a priori of $\delta D$ and the constraint for $N_2O$, $CH_4$, and $HNO_3$. This difference between the two versions is actually not noticeable by the user and the here provided report on version 3.2.1 data is also valid for version 3.3.0 data, which will also soon be made available for the public.

## 3.2 Variables

There are three different categories of variables. The first category consists of variables that contain information resulting from the EUMETSAT L2 PPF (product processing facility) retrieval. They can be identified by the prefix `eumetsat_` in their names. A second category consists of variables that contain information from the MUSICA IASI retrieval. Here the prefix in the name is `musica_`. The third category encompasses all other variables and their names have no specific prefix.

The EUMETSAT L2 retrieval variables are flags (mainly for cloud coverage, see Sect.4.1, surface conditions, and EUMETSAT retrieval quality) and the EUMETSAT L2 retrieval output of $H_2O$. The variables belonging to the third category are supporting data and inform about the sensors' viewing geometry, observation time, measured radiances, climatological tropopause altitude, and surface emissivity. Although our MUSICA IASI retrieval uses the EUMETSAT L2 PPF version 6 land surface emissivity, the emissivity variables are assigned to the category of supporting data, because for older observations





where no L2 PPF version 6 is available we use the surface emissivity climatology from IREMIS (Seemann et al., 2008) and over water we always use the values reported by Masuda et al. (1988). The large majority of variables are MUSICA IASI variables. These variables document the MUSICA IASI retrieval settings (like the a priori states and constraints, see Sects. 4.4 to 4.6), provide the MUSICA IASI retrieval products (retrieved trace gas profiles, Sect. 5.1), and characterise these products (averaging kernels, estimated errors, Sect. 5.2). For variables that refer to a specific retrieval product, a corresponding syllable

is embedded into the respective variable names: `_wv_` and `_wvp_` stands for water vapour isotopologues and water vapour isotopologue proxies, respectively, `_ghg_` for the greenhouse gases $N_2O$ and $CH_4$, `_hno3_` for $HNO_3$, and `_at_` for the atmospheric temperature.

The water vapour and greenhouse gas variables (`_wv_`, `_wvp_`, and `_ghg_`) contain information on two species, which can be identified by the value of the dimension `musica_species_id`. For `_wv_` these are the species $H_2O$ and HDO, for

`_wvp_` the water vapour proxy species (see Sect. 4.4.2), and for `_ghg_` the species $N_2O$ and $CH_4$, respectively.

## 4 MUSICA IASI retrieval set up

In this section the principle setup of the MUSICA IASI retrieval is presented. We discuss our filtering before processing, the used retrieval algorithm, the measurement state (spectral region), the atmospheric state that is retrieved in an optimal estimation sense, and the used a priori information and the applied constraints. A detailed explanation of these settings ensures the full

reproducibility of the data and is also important in the context of data reusage (see examples given in Sect. 8).

### 4.1 Data selection prior to processing

We focus on the processing of IASI data for which EUMETSAT L2 data files of PPF version 6.0 or later are available. For former data versions not all of the subsequently discussed L2 PPF variables are available. Furthermore, we found that there are several modifications made within version 4 and 5 that significantly affect the stability of our MUSICA IASI retrieval output

(see discussion in García et al., 2018). EUMETSAT L2 PPF verion 6 data are available from October 2014 onward, so we focus our processing on IASI observations made from October 2014 onward.

In addition, the MUSICA IASI retrievals are currently restricted to cloud-free scenarios. The selection of cloud-free conditions is made by means of the EUMETSAT L2 PPF cloudiness assessment summary flag variable (called `flag_cldnes` in the EUMETSAT L2 netcdf data files). We only process IASI observation with this flag having the value 1 (the IASI Instrumen-

tal Field Of View, IOFV, is clear) or 2 (the IASI IFOV is processed as cloud-free but small cloud contamination possible). This requirement for cloud free scenarios removes more than 2/3 of all available IASI observations.

Furthermore, we require EUMETSAT L2 PPF temperture profiles being generated by the EUMETSAT L2 PPF optimal estimation retrieval scheme. For this porpuse we use the EUMETSAT L2 PPF variable `flag_itconv`. We only process data with this flag having value 3 (the minimisation did not converge, sounding accepted) or 5 (the minimisation converged,

sounding accepted).

Earth System
Open Access   Science   Discussions
Data

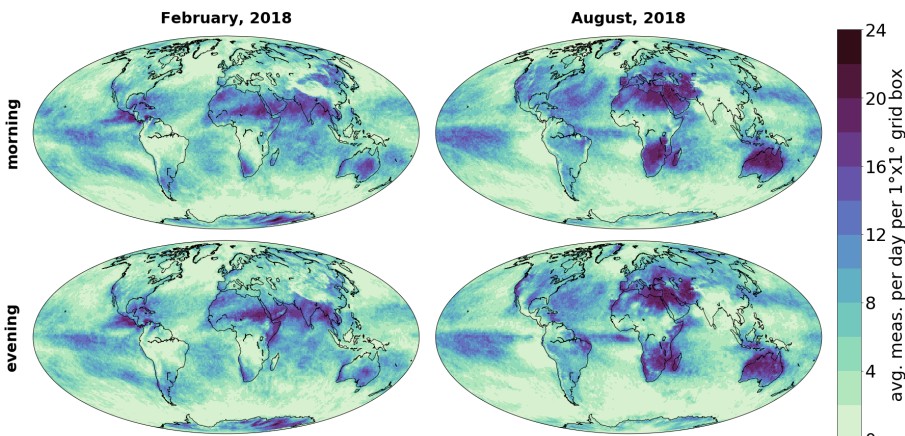

**Figure 2.** Monthly averaged number of the IASI observations per $1° \times 1°$ box that passed our selection criteria prior to the MUSICA IASI processing, for February and August 2018 and for local morning and evening overpasses.

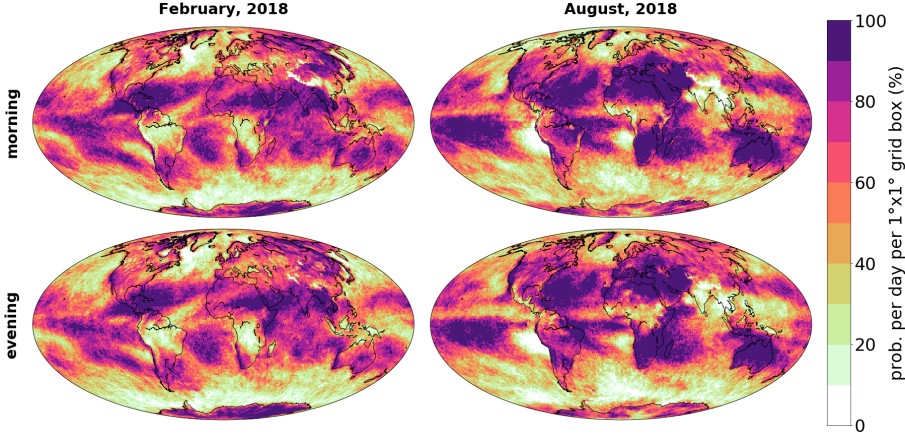

**Figure 3.** Similar as Fig.2, but for probability of having at least one valid IASI observation (measurement that passed the cloud filter and EUMETSAT quality checks) per overpass and $1° \times 1°$ box.

Figure 2 gives a climatological overview on the amount of IASI data that remain after the aforementioned preselection. The maps largely reflect the cloud cover conditions. A very large amount of IASI data passed our selection criteria in the suptropical regions, where cloud-free condition generally prevail. In the North Atlantic storm track region, the South American and South African tropics, and the Southern Polar Oceans the sky is generally cloudy in February leading to a low number of
IASI observations that passed our selection criteria. In August we can clearly identify the Asian and West African monsoon region as an area with increased coud coverage and consequently less MUSICA IASI processed data.

Figure 3 is similar to Fig. 2, but instead of showing the total number of observations that fall within a $1° \times 1°$ box it depicts the probability of having at least one observation per overpass in a $1° \times 1°$ box. In both figures we observe very similar structures.





## 4.2 The retrieval algorithm

We use the thermal nadir retrieval algorithm PROFFIT-nadir (Schneider and Hase, 2011; Wiegele et al., 2014). It is an extension of the PROFFIT algorithm (PROFile Fit, Hase et al., 2004) used since many years by the ground-based infrared remote sensing community (Kohlhepp et al., 2012; Schneider et al., 2012). This extension has been made in support of the IASI retrieval development during the project MUSICA. The algorithm consists of a line by line radiative transfer code PROFFWD (Hase et al., 2004; Schneider and Hase, 2009) and can consider Voigt as well as non-Voigt line shapes (Gamache et al., 2014) and

the water continuum signatures according to the model MT_CKD v2.5.2 (Delamere et al., 2010; Payne et al., 2011; Mlawer et al., 2012). For the MUSICA IASI processing we use the water continuum model MT_CKD v2.5.2 and for all trace gases a Voigt line shape model and the spectroscopic line parameters according to the HITRAN2016 molecular spectroscopic database (Gordon et al., 2017). However, we increase the line intensity parameter for all HDO lines by +10%, in order to correct for the bias observed between MUSICA IASI $\delta$D retrievals and respective aircraft-based in-situ profile data (Schneider et al., 2015).

For the inversion calculations PROFFIT-nadir offers options that are essential for water vapour isotopologue retrievals. These are the options for logarithmic scale retrievals and for setting up a cross constraint between different atmospheric species (see also Sect. 4.4.2). The theoretical basics for atmospheric trace gas retrievals are provided in Appendix A.

## 4.3 The analysed spectral region

The retrieval works with the radiances measured in the spectral region between $1190\,\mathrm{cm}^{-1}$ and $1400\,\mathrm{cm}^{-1}$. The respective

radiance values are the elements of the MUSICA IASI measurement state vector referred to as $\boldsymbol{y}$ in Appendix A. Figure 4 depicts measured and simulated radiances as well as a large variety of different Jacobians for a typical mid-latitudinal summer observation over land. Please note the different radiance scale for measurement and simulation, on the one hand, and residuals and Jacobians, on the other hand.

We show trace gas Jacobians for an uniform increase of the trace gases throughout the whole atmosphere: 100% for $H_2O$ and

HDO, 10% for $N_2O$ and $CH_4$, 50% for $HNO_3$. The respective values are reasonable approximations to the typical atmospheric variabilities of these trace gases. We see that the measured radiances are most strongly affected by the water isotopologues. The variations of $N_2O$ and $CH_4$ are also recognizable (larger than the spectral residuals, i.e. the difference between measured and simulated radiances). The Jacobians of $HNO_3$ are very close to the noise level.

The atmospheric temperature Jacobians are depicted for a uniform $2\,\mathrm{K}$ temperature increase over three different layers:

surface - 2 km a.s.l., 2 - 6 km a.s.l., and 6 - 12 km a.s.l. Atmospheric temperature variations close to the surface affect mainly the radiances below $1300\,\mathrm{cm}^{-1}$ and variations at higher altitudes mainly the radiances above $1300\,\mathrm{cm}^{-1}$. In Fig. 4 we depict the Jacobians for $2\,\mathrm{K}$, because this is a resonable approximation of the uncertainty in the EUMETSAT L2 PPF temperatures (August et al., 2012).

The Jacobian for surface emissivity and temperature reveal that surface properties hardly affect the radiances above $1250\,\mathrm{cm}^{-1}$,

but have a strong impact below $1250\,\mathrm{cm}^{-1}$. We calcuate the emissivity Jacobias for a $-2\%$ change of the emissivity independently above an below $1270\,^{-1}$, which is a typical uncertainty of emissivity judging from its dependency on viewing angle and



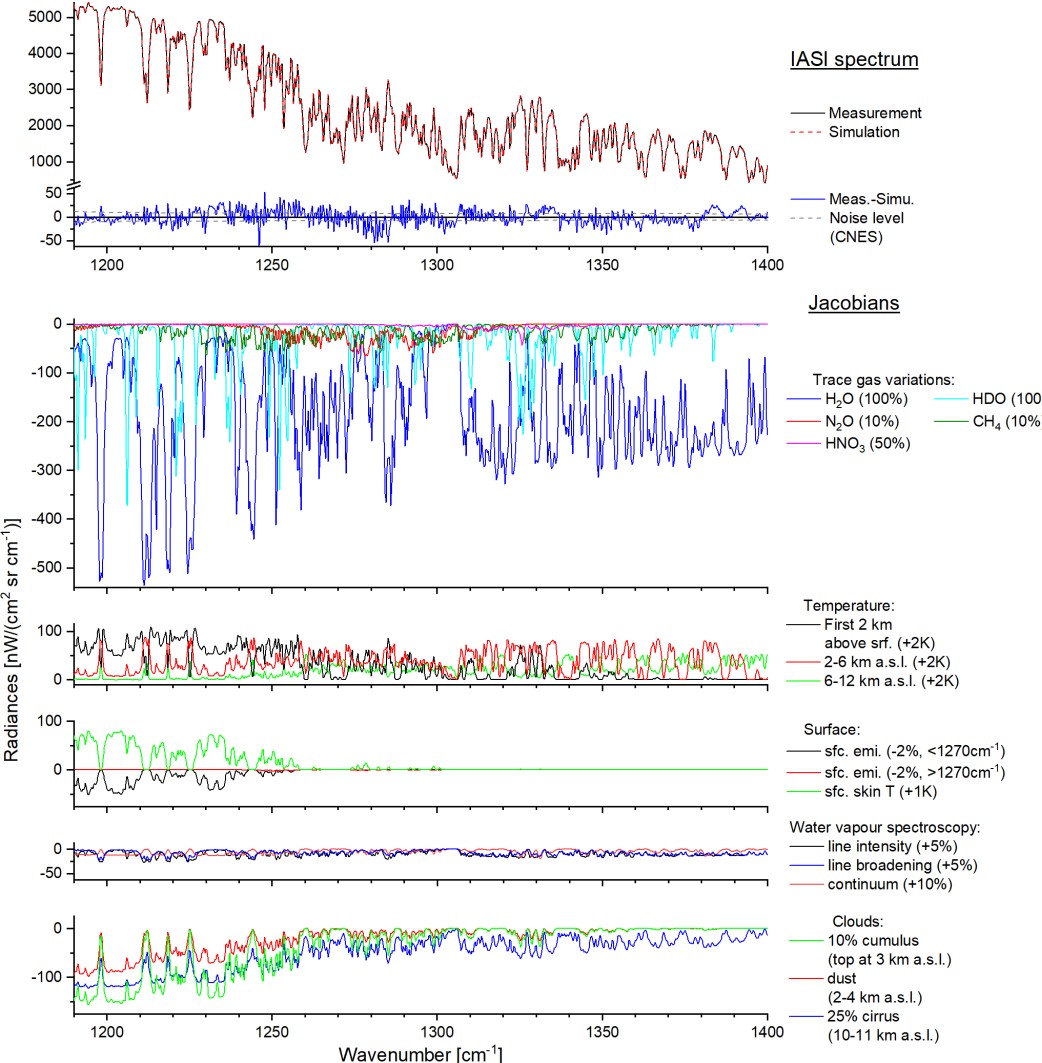

**Figure 4.** Example for a spectrum, residual (difference between measured and simulated spectrum) and different spectral responses (Jacobians) for an observation over the mid-latitudinal site of Lindenberg on 30. Aug. 2008 (satellite nadir angle: 43.7°; surface skin temperature: 292.3 K; precipitable water vapour: 31.8 mm).

wind speed over ocean (Masuda et al., 1988) and small scale inhomogeneities; however, this uncertainty might be significantly higher over arid areas (Seemann et al., 2008).

Concerning spectroscopy, the Jacobians calculated for the typical uncertainty range of spectroscopic parameters are relatively 230 small. In Fig. 4 we show the Jacobians for consistent +5% changes of the line intensity and pressure broadening parameters of all water vapour isotoplogues, which is in reasonable agreement with the uncertainty values given by HITRAN (Gordon





et al., 2017). The Jacobians shown are for a water continuum that is by 10% larger than the continuum according to the model
MT_CKD v2.5.2 (Delamere et al., 2010; Payne et al., 2011; Mlawer et al., 2012).

The bottom panel of Fig. 4 depicts the impact of clouds on the radiances. The thermal nadir radiance when observing over an
opaque cloud can be calculate by defining the cloud top instead of the surface as the thermal background. Cirrus and mineral
dust clouds are not opaque and we have to consider partial attenuation by the cloud particles. We calculate the attenuated
radiances using forward model calculations from KOPRA (Karlsruhe Optimized and Precise Radiative transfer Algorithm;
Stiller, 2000) and consider single scattering. The frequency dependency of the extinction cross sections, the single scattering
albedo, and the scattering phase functions of the clouds are calculated from OPAC v4.0b (Optical Properties of Aerosol and
Clouds; Hess et al., 1998; Koepke et al., 2015). For cirrus clouds we assume the particle composition as given by OPAC's
"Cirrus 3" ice cloud example (see Table 1b in Hess et al., 1998) and for mineral dust clouds a particle composition according
to OPAC's "Desert" aerosol composition example (see Table 4 in Hess et al., 1998). The shown Jacobians are for 10% cumulus
cloud coverage with the cloud top at 3 km, a homogeneous dust cloud between 2 and 4 km, and 25% cirrus cloud coverage
between 10 and 11 km. These are relatively weak clouds and we assume that they might occasionally not correctly be identified
by the EUMETSAT L2 cloud screening algorithm. Because the respective Jacobians are significantly above the noise level,
these unrecognised clouds can have an important impact on the retrieval.

A comprehensive set of different Jacobians is provided with the extended output data file for the 74 examplatory observation
at an Arctic, mid-latitudinal and tropical site.

### 4.4  The state vector

In this Section we discuss the MUSICA IASI state vector, which is referred to as $x$ in Appendix A.

#### 4.4.1  Components of the state vector

We retrieve vertical profiles of the trace gases $H_2O$, HDO, $N_2O$, $CH_4$, $HNO_3$, and of atmospheric temperature. For all these
profile retrievals we use constraints (for more details see Sect. 4.6). In addition we fit the surface skin temperature and the
spectral frequency scale without any constraint. We discretise the profiles on atmospheric levels between the surface and the
top of the atmosphere (which we set at 56 km). The grid is relatively fine in the lower troposphere ($\approx 400$ m) and increases
in the stratosphere to above 5 km. The number of atmospheric levels ($nal$) depends on the surface altitude. For instance, for a
surface altitude at sea level (0 m a.s.l.) $nal = 28$ and for a surface altitude of 4000 m a.s.l. $nal = 21$. Consequently, the state
vector for an observation with surface altitude at sea level has a length of $6 \times 28 + 2 = 170$.

#### 4.4.2  Water vapour isotopologue proxies

The water vapour isotoplogues $H_2O$ and HDO vary largely in parallel. The information that HDO actually adds to $H_2O$ lies
in the value of the HDO/$H_2O$ ratio. This ratio is typically expressed as $\delta D = \frac{H_2O/HDO}{VSMOW} - 1$, with $VSMOW = 3.1152 \times 10^{-4}$
(VSMOW: Vienna Standard Mean Ocean Water). In Schneider et al. (2006b) the logarithmic scale difference between $H_2O$





and HDO has been introduced as a good proxy for $\delta$D and Schneider et al. (2012) showed that by a transformation between the state {H$_2$O,HDO} – needed for the radiative transfer calculations – and the proxy state {$\frac{1}{2}$(ln[H2O]+ln[HDO]),$\frac{1}{2}$(ln[HDO]−

ln[H2O])} – where we can formulate the correct constraints – the climatologically expected variability of the atmospheric state can be described correctly.

First we have to transfer the associated mixing ratio entries in the state vector to a logarithmic scale. This means that all the derivatives provided by the radiative transfer caluulations have to be transferred from the linear scale to the logarithmic scale by setting $\partial x = x \partial \ln[x]$. For highly variable trace gases logarithmic scale retrievals are advantageous, because they allow

considering the correct a priori statistics (log-normal instead of normal distributions, Hase et al., 2004; Schneider et al., 2006a). For trace gases with weak variability but still detectable spectral signatures, the statistics in logarithmic and linear scale become very similar, so logarithmic scale retrievals have no apparent disadvantage with respect to linear scale retrievals, instead they offer unique possibilities as outlined in the following. In logarithmic scale the water vapour isotopologue state can be expressed in the basis of {ln[H$_2$O], ln[HDO]} or in the basis of the proxy state {$\frac{1}{2}$(ln[H$_2$O]+ln[HDO]),(ln[HDO]−ln[H$_2$O])}. Both

expressions are equally valid. Each basis has the dimension $(2 \times nal)$. In the following the full water vapour isotopologue state vector expressed in the {ln[H$_2$O], ln[HDO]} basis and the {$\frac{1}{2}$(ln[H$_2$O]+ln[HDO]),(ln[HDO]−ln[H$_2$O])} proxy basis will be referred to as $\boldsymbol{x}$ and $\boldsymbol{x}'$, respectively. The basis transformation can be achieved by operator $\mathbf{P}$:

$$\mathbf{P} = \begin{pmatrix} \frac{1}{2}\mathbf{I} & \frac{1}{2}\mathbf{I} \\ -\mathbf{I} & \mathbf{I} \end{pmatrix} \tag{1}$$

Here the four matrix blocks have the dimension $(nal \times nal)$, $\mathbf{I}$ stands for an identity matrix and the state vectors $\boldsymbol{x}$ and $\boldsymbol{x}'$

are related by:

$$\boldsymbol{x}' = \mathbf{P}\boldsymbol{x} \tag{2}$$

Similarly logarithmic scale covariance matrices can be expressed in the two basis systems and the respective matrices $\mathbf{S}$ and $\mathbf{S}'$ are related by:

$$\mathbf{S}' = \mathbf{P}\mathbf{S}\mathbf{P}^T \tag{3}$$

and respective averaging kernel matrices $\mathbf{A}$ and $\mathbf{A}'$ are related by:

$$\mathbf{A}' = \mathbf{P}\mathbf{A}\mathbf{P}^{-1} \tag{4}$$

In contrast to H$_2$O and HDO, H$_2$O and $\delta$D vary to a large extent independently, and we can easily set up the constraint matrix $\mathbf{R}'$ for the proxy basis {$\frac{1}{2}$(ln[H$_2$O]+ln[HDO]),(ln[HDO]−ln[H$_2$O])}:

$$\mathbf{R}' = \begin{pmatrix} \mathbf{R}_{\mathrm{H20}} & 0 \\ 0 & \mathbf{R}_{\delta\mathrm{D}} \end{pmatrix}. \tag{5}$$

Back transformation to the {ln[H$_2$O], ln[HDO]} basis reveals automatically the strong cross constraints between H$_2$O and HDO:

$$\mathbf{R} = \mathbf{P}^{-1}\mathbf{R}'\mathbf{P}^{-T} = \begin{pmatrix} \frac{1}{2}\mathbf{R}_{\mathrm{H20}} + \frac{1}{2}\mathbf{R}_{\delta\mathrm{D}} & \frac{1}{2}\mathbf{R}_{\mathrm{H20}} - \frac{1}{2}\mathbf{R}_{\delta\mathrm{D}} \\ \frac{1}{2}\mathbf{R}_{\mathrm{H20}} - \frac{1}{2}\mathbf{R}_{\delta\mathrm{D}} & \frac{1}{2}\mathbf{R}_{\mathrm{H20}} + \frac{1}{2}\mathbf{R}_{\delta\mathrm{D}} \end{pmatrix}. \tag{6}$$



For more details on the utility of the water vapour isotopologue proxy state please refer to Schneider et al. (2012) and Barthlott et al. (2017).

### 4.4.3 Summary

The atmospheric state variables that are independently constrained during the MUSICA IASI processing are the vertical profiles of the water vapour isotoplogue proxies $H_2O$ and $\delta D$, and the vertical profiles of $N_2O$, $CH_4$, $HNO_3$, and atmospheric temperature. For all the trace gases (not only for the water vapour isotoplogues) the retrieval works with the state variables in a logarithmic scale. For atmospheric temperature a linear scale is used. Surface skin temperature and the spectral frequency shift are also components of the state vector; however, they are not constrained during the retrieval procedure.

The variables `musica_wv_apriori` and `musica_wv` provide the a priori assumed and the retrieved values of $H_2O$ and HDO, respectively (see also Sect.4.5 and 5.1). The output is given in ppmv and normalised with respect to the naturally occurring isotopologue abundance. In this context, $\delta D$ is calculated from the content of these variables as $\delta D = 1000 \left( \frac{\text{HDO}}{\text{H}_2\text{O}} - 1 \right)$. Information about $H_2O$ and $\delta D$ related to differentials (constraints, averaging kernels, kernel metrics, or uncertainties) are generally provided in the proxy states (variables with the syllable `_wvp_`).

### 4.5 A priori states

The reference for the a priori data used for the MUSICA IASI trace gas retrievals is the CESM1/WACCM (Community Earth System Model version 1 / Whole Atmosphere Community Climate Model) monthly output of the 1979-2014 time period. The CESM1/WACCM is a coupled chemistry climate model from the Earth's surface to the lower thermosphere (Marsh et al., 2013). The horizontal resolution is $1.9°$ latitude x $2.5°$ longitude. The vertical resolution in the lower stratosphere ranges from $1.2\,\text{km}$ near the tropopause to about $2\,\text{km}$ near the stratopause; in the mesosphere and thermosphere the vertical resolution is about $3\,\text{km}$. Simulations used for generating the MUSICA IASI a priori data are based on the International Global Atmospheric Chemistry / Stratosphere-troposphere Processes And their role in Climate (IGAC/SPARC) Chemistry Climate Model Initiative (CCMI, Morgenstern et al., 2017). From the surface to $50\,\text{km}$ the meteorological fields are nudged towards meteorological analysis taken from the National Aeronautics and Space Administration (NASA) Global Modeling and Assimilation Office (GMAO) Modern-Era Retrospective Analysis for Research and Applications (MERRA, Rienecker et al., 2011) and above $60\,\text{km}$ the model meteorological fields are fully interactive, with a linear transition in between (details about the nudging approach are described in Kunz et al., 2011).

For the MUSICA IASI a priori profiles of $H_2O$, $N_2O$, $CH_4$ and $HNO_3$ we consider a mean latitudinal dependence, sesonal cycles, and long-term evolution. Therefore, the a priori data are constructed by means of a low dimensional multi-regression fit on the CESM1/WACCM data independently for each vertical grid level. We fit an annual cycle with the two frequencies 1/year and 2/year, and for the long-term baseline we fit a second order polynom. The fits are performed individually for 15 equidistant latitudinal bands between $90°$S and $90°$N. In order to capture the yearly anomalies in $N_2O$ and $CH_4$ a priori data, we use the Mauna Loa Global Atmospheric Watch yearly mean data records for a correction of the WACCM parameterized time series (for more details on this correction procedure see Barthlott et al., 2015). We also use the temperature lapse rate

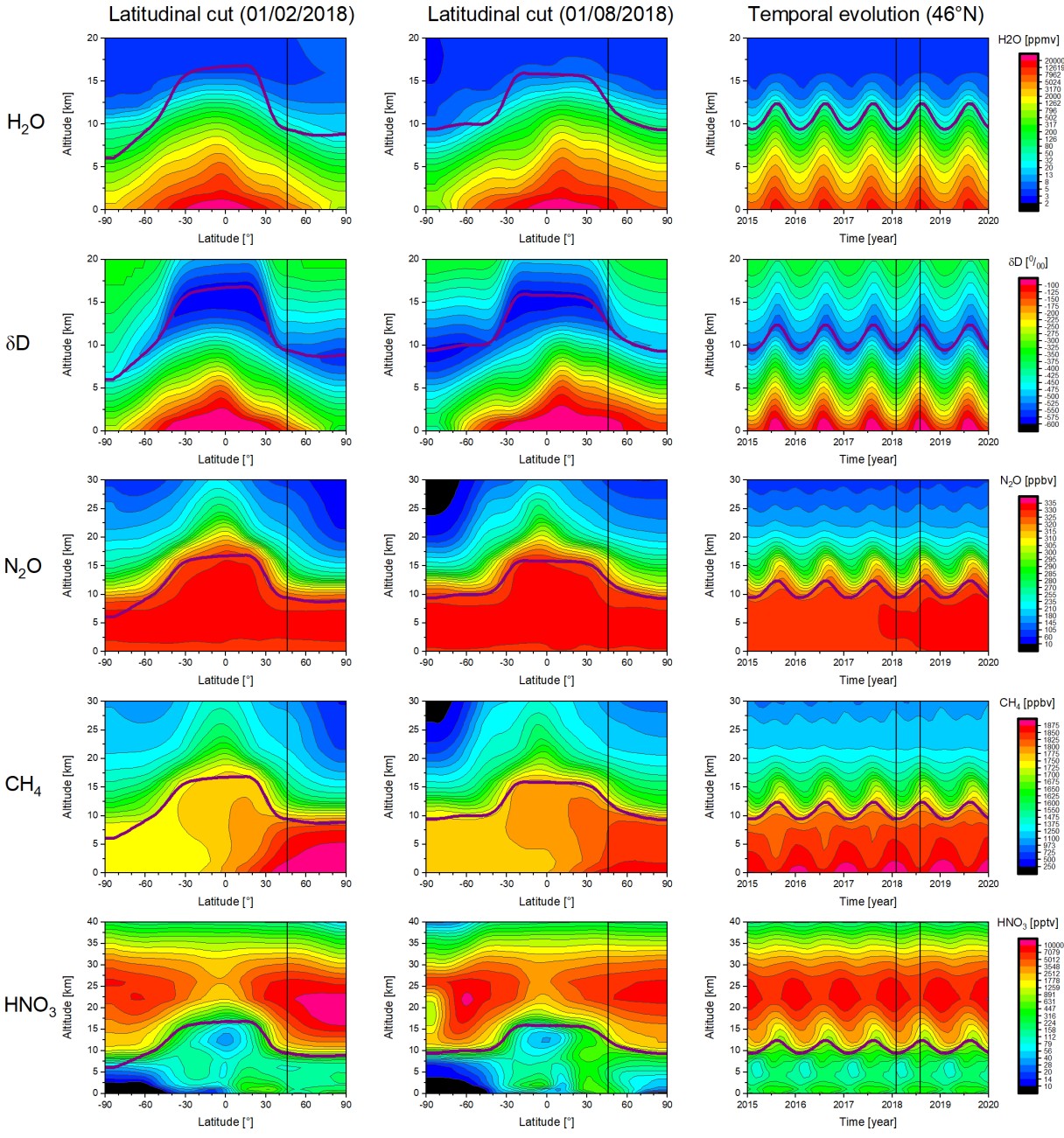

**Figure 5.** Overview on time and latitude dependent a priori information (source: WACCM model simulations) used for the MUSICA IASI retrieval for all targeted atmospheric species (H$_2$O, $\delta$D, N$_2$O, CH$_4$, and HNO$_3$) and for the tropopause altitude (depicted as violett thick solid line). Shown are latitudinal cross sections for 01/02/2018 and 01/08/2018 and the temporal evolution at 46°N for the 2015-2020 time period. Please note the non-uniform y-axes scales.





tropopause – according to the definition of the World Meteorological Organisation – from WACCM and construct a latitudinal dependent tropopause altitude by fitting a seasonal cycle and a constant baseline (no long-term dependency) and assume a transition zone between troposphere and stratosphere with a vertical extension of $12.5\,\mathrm{km}$. The MUSICA IASI $\delta$D a priori profiles between ground and the tropopause altitude are constructed from the $H_2O$ a priori profiles by using a single global

relation between tropospheric $H_2O$ concentration and $\delta$D values. This relation has been determined from simultaneous $H_2O$ and $\delta$D measurements made by high precision in-situ instruments at different ground stations located in the mid-latitudes and the subtropics and between $100\,\mathrm{m}$ and $3650\,\mathrm{m}$ a.s.l. (González et al., 2016; Christner et al., 2018) and by aircraft-based in-situ measurements made between sea surface and about $7000\,\mathrm{m}$ a.s.l. (Dyroff et al., 2015). Above the troposphere we smoothly connect the tropospheric $\delta$D values with the typical stratospheric $\delta$D value of $-350‰$.

Figure 5 depicts the MUSICA IASI a priori data derived from WACCM. It shows latitudinal cross sections for a northern hemispheric winter and summer day as well as the temporal evolution between 2014 and 2020 at a mid-latitudinal site. The $H_2O$ and $\delta$D a priori have strong latitudinal gradients and also a marked seasonal cycle. For $\delta$D the lowest values are in the neighbourhood of the tropopause altitude (depicted as violet thick line). The a prioris of $N_2O$ and $CH_4$ have a strong latitudinal and seasonal variability in the tropopause region. $CH_4$ has a strong tropospheric latitudinal gradient and seasonal cycle in

the troposphere, whereas the tropospheric $N_2O$ variability is rather small. The $HNO_3$ a priori has a maximum in the lower stratosphere $(20-25\,\mathrm{km})$ with highest values at higher latitudes.

The a priori trace gas profiles are provided in the variables `musica_wv_apriori` ($H_2O$ and HDO with species index 1 and 2, respectively), `musica_ghg_apriori` ($N_2O$ and $CH_4$ with species index 1 and 2, respectively), and `musica_hno3_apriori` ($HNO_3$). The unit is ppmv.

As a priori for the atmospheric and the surface temperatures we use the EUMETSAT L2 PPF atmospheric temperature output. These data are provided in Kelvin and in the variables `musica_at_apriori` and `musica_st_apriori`, for atmospheric temperature and surface temperature, respectively.

## 4.6 A priori covariances and constraints

We set up simplified a priori covariance matrices by means of two parameters. The first parameter are the altitude dependent
amplitudes of the variability ($v_{\mathrm{amp},i}$, with $i$ indexing the $i$th altitude level). For the trace gases we work with the relative variability, i.e. with the variability on the logarithmic scale. For atmospheric temperatures the variability is given in Kelvin. The second parameter are the altitude dependent vertical correlation lengths ($\sigma_{\mathrm{cl},i}$, for considering correlated variations between different altitudes). The elements of the a priori covariance matrix ($\mathbf{S_a}$) are then calculated as:

$$S_{\mathrm{a}i,j} = v_{\mathrm{amp},i} v_{\mathrm{amp},j} \exp{-\frac{(z_i - z_j)^2}{2\sigma_{\mathrm{cl},i}\sigma_{\mathrm{cl},j}}}, \tag{7}$$

with $z_i$ being the altitude at the $i$th altitude level.

The values $v_{\mathrm{amp},i}$ and $\sigma_{\mathrm{cl},i}$ are oriented to the typical covariances of in-situ observations made from ground (e.g. González et al., 2016; Gomez-Pelaez et al., 2019), aircraft (e.g. Wofsy, 2011; Dyroff et al., 2015), or balloon (e.g. Karion et al., 2010; Dirksen et al., 2014) and also aligned to the vertical dependency of the monthly mean covariances we obtain from the WACCM




simulations. For $v_{\mathrm{amp},i}$ of $\delta$D we use in addition the isotopologue enabled version of the Laboratoire de Météorologie Dy-
namique (LMD) general circulation model as a reference (Risi et al., 2010; Lacour et al., 2012). For atmospheric temperature
we use the uncertainty in the EUMETSAT L2 atmospheric temperature as reference (August et al., 2012). Generally, we
classify three different altitude regions with specific vertical dependencies in the values of $v_{\mathrm{amp},i}$ and $\sigma_{\mathrm{cl},i}$: the troposphere
(below the climatological tropopause altitude as depicted in Fig. 5), the stratosphere (starting 12.5 km above the climatological
tropopause altitude), and the transition region between troposphere and stratosphere.

The values of $v_{\mathrm{amp},i}$ are specific for each trace gas and for the atmospheric temperature and they are provided in the MUSICA
IASI standard output files in the variables having the suffix `_apriori_amp`. As a simplification we use the same values of
$\sigma_{\mathrm{cl},i}$ for all trace gases and for the atmospheric temperature. These values are provided in the MUSICA IASI output files as the
variable `musica_apriori_cl`.

As the constraint of the retrieval we use an approximation of the inverse of the covariance matrix. For this purpose the
constraint matrix R is constructed as a sum of a diagonal constraint, and first and second order Tikhonov-type regularisation
matrices (Tikhonov, 1963):

$$\mathbf{R} = (\alpha_{\mathbf{0}}\mathbf{L_0})^T \alpha_{\mathbf{0}}\mathbf{L_0} + (\alpha_{\mathbf{1}}\mathbf{L_1})^T \alpha_{\mathbf{1}}\mathbf{L_1} + (\alpha_{\mathbf{2}}\mathbf{L_2})^T \alpha_{\mathbf{2}}\mathbf{L_2}, \tag{8}$$

with:

$$\mathbf{L_0} = \begin{pmatrix} 1 & 0 & 0 & \cdots & 0 \\ 0 & 1 & 0 & \cdots & 0 \\ 0 & 0 & 1 & \cdots & 0 \\ \vdots & \vdots & \vdots & \ddots & \vdots \\ 0 & 0 & 0 & \cdots & 1 \end{pmatrix}, \tag{9}$$

$$\mathbf{L_1} = \begin{pmatrix} 1 & -1 & 0 & \cdots & 0 \\ 0 & 1 & -1 & \cdots & 0 \\ \vdots & \vdots & \ddots & \ddots & \vdots \\ 0 & \cdots & 0 & 1 & -1 \end{pmatrix}, \tag{10}$$

and

$$\mathbf{L_2} = \begin{pmatrix} 1 & -2 & 1 & \cdots & 0 \\ \vdots & \ddots & \ddots & \ddots & \vdots \\ 0 & \cdots & 1 & -2 & 1 \end{pmatrix}. \tag{11}$$

The diagonal elements of the diagonal matrices $\alpha_{\mathbf{0}}$, $\alpha_{\mathbf{1}}$, and $\alpha_{\mathbf{2}}$ are the inverse of the absolute variabilities and the variabilities
of the first and the second vertical derivatives of the profiles. These values can be calculated from the elements of the a priori
matrix ($\mathbf{S_a}$) as follows:

$$\alpha_{0i,i} = \left(\sqrt{\mathrm{S}_{\mathrm{a}i,i}}\right)^{-1}, \tag{12}$$





$$\alpha_{1_{i,i}} = \left( \sqrt{S_{a_{i,i}} + S_{a_{i+1,i+1}} - 2S_{a_{i,i+1}}} \right)^{-1}, \tag{13}$$

$$\alpha_{2_{i,i}} = \left( \sqrt{S_{a_{i,i}} + 4S_{a_{i+1,i+1}} + S_{a_{i+2,i+2}} - \left( 4S_{a_{i,i+1}} - 2S_{a_{i,i+2}} + 4S_{a_{i+1,i+2}} \right)} \right)^{-1}. \tag{14}$$

Starting the retrievals with the constarint matrix $\mathbf{R} \approx \mathbf{S_a}^{-1}$ optimises the computational efficiency of the retrieval processes, because according to Eqs. (A4) and (A5) the retrieval calculations work with $\mathbf{S_a}^{-1}$. Furthermore, calculating the inversion of $\mathbf{S_a}$ approximatively as the sum of diagonal constraint, and first and second order Tikhonov-type regularisation matrices offers the possibility of tuning the constraint according to specific user requirements with respect to smoothness or absolute deviations (e.g. Steck, 2002; Diekmann et al., 2021).

For the greenhouse gases ($N_2O$ and $CH_4$) and $HNO_3$ we constrain with respect to the absolute values of the profiles and the first derivative of the profile, i.e. we do not consider the term $(\alpha_2 \mathbf{L_2})^T \alpha_2 \mathbf{L_2}$ of Eq. (8). In case of the water vapour isotopologue proxies and the atmospheric temperature we additionally constrain with respect to the second derivative of the profile, i.e. we consider all terms of Eq. (8). Please note that for the trace gases the constraints work on the logarithmic scale and for the atmospheric temperature on the linear scale.

Because $HNO_3$ has only very weak spectroscopic signatures in the analysed spectral region (see Fig. 4) we loosen the absolute constraint and at the same time strengthen the constraint with respect to the first vertical derivate: $\alpha_0$ and $\alpha_1$ are calculated from an $\mathbf{S_a}$ constructed with the values of $v_{\mathrm{amp},i}$ increased by a factor of 1.5 and with the values of $\sigma_{\mathrm{cl},i}$ increased by a factor of 2. Similarly and in order to avoid a negative impact of an underconstrained retrieval of the temperature profile on the trace gas products (e.g. artificial oscillatory features), we strengthen the atmospheric temperature constraint: $\alpha_0$, $\alpha_1$ and $\alpha_2$ are calculated from an $\mathbf{S_a}$ constructed with the values of $v_{\mathrm{amp},i}$ decreased by a factor of 0.5.

The digonal entries of the diagonal matrices $\alpha_0$, $\alpha_1$, and $\alpha_2$ contain all information about the actual constraints used by the retrieval. They are provided in the MUSICA output files for each individual retrieval and for the different trace gases and the atmospheric temperature as the variables with the suffix `_reg`. For the trace gases these vector elements are depicted in Fig. 6 for a northern hemispheric summer in the tropics, mid-latitudes and polar regions. The dotted lines indicate the climatological tropopause and the altitude $12.5\,\mathrm{km}$ above this tropopause (transition zone between troposphere and stratosphere).

## 5 MUSICA IASI retrieval output

In this sections we describe the variables that inform about the retrieval traget products (vertical trace gas profiles) and the characteristics of these products (averaging kernels and errors). A detailed explanation of these data supports their interoperability and is also important in the context of data reusage (see examples given in Sect. 8).

### 5.1 Trace gas profiles and temperatures

The retrieved trace gas profiles are provided in the variables `musica_wv` ($H_2O$ and HDO with species index 1 and 2, respectively), `musica_ghg` ($N_2O$ and $CH_4$ with species index 1 and 2, respectively), and `musica_hno3` ($HNO_3$). The unit is

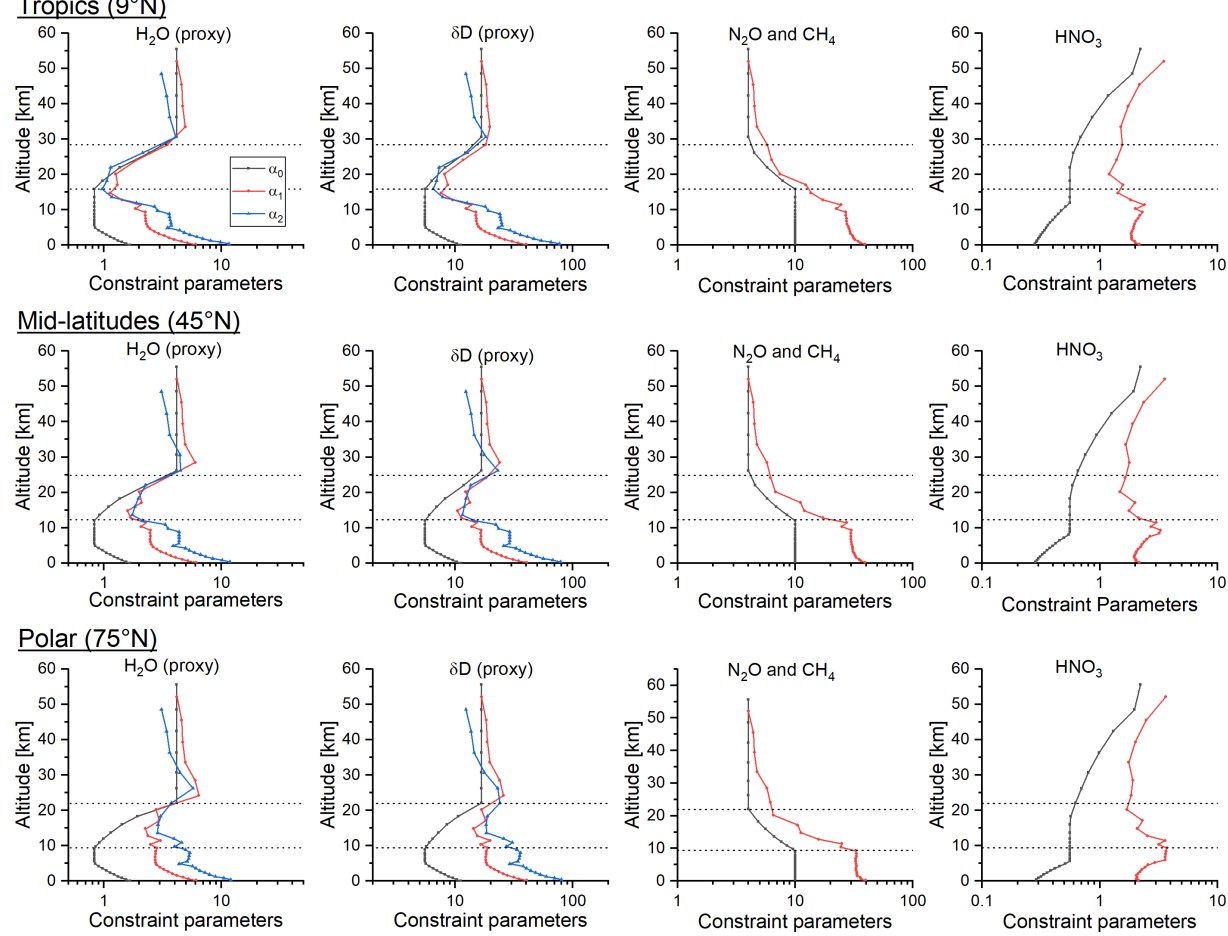

**Figure 6.** Vertical profiles of the constraint parameter $\alpha_0$, $\alpha_1$, and $\alpha_2$ for the retrieval of all targeted atmospheric species ($H_2O$, $\delta D$, $N_2O$, $CH_4$, and $HNO_3$). The parameters are latitudinal and time dependent. Shown are examples for 8. July 2018 for the tropics ($9°N$), mid-latitudes ($45°$), and polar regions ($75°$). The area between the dotted lines indicates the transition between troposphere and stratosphere.

415    ppmv. The retrieved atmospheric temperature is provided in the variable `musica_at` and the retrieved surface temperature in the variable `musica_st`. The unit is Kelvin.

In order to provide a brief insight into the data diversity, Figure 7 gives examples with a apriori and retrieved trace gas profiles for an observation on 30. Aug. 2008 over Lindenberg (53°N). The profile data represent 28 altitude levels and are provided with detailed information of their sensitivity, vertical representativeness, and errors (see following subsections).

**5.2   Characteristics of retrieved products**

For a limited number of retrievals we provide an extended netcdf output file (see Sect. 3.1). The extended output file contains the same variables as the standard output files and in addition the full averaging kernels and a large set of Jacobians together



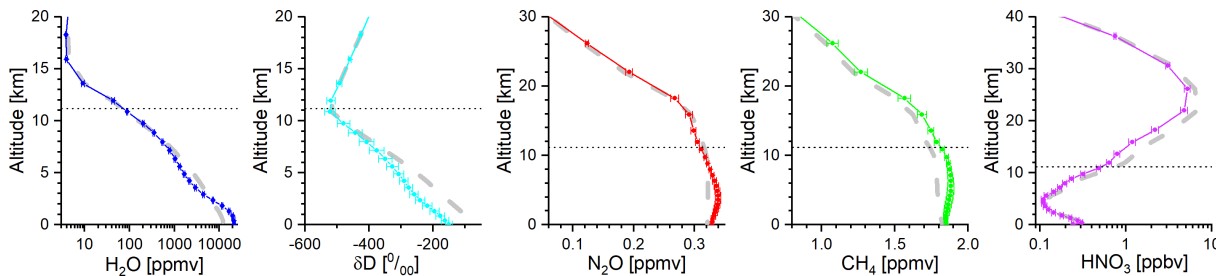

**Figure 7.** Example of vertical profiles of the targeted atmospheric trace gas products ($H_2O$, $\delta D$, $N_2O$, $CH_4$, and $HNO_3$): thick grey dashed lines represent the a priori assumption adn the coloured line with symbols and error bars are the retrieved values and the root-square-sum of the fit noise error and the estimated temperature error. The data are for the 30. Aug. 2008 example observation over Lindenberg (53°N) used for Fig. 4.

with gain matrices. The latter allows the calculation of full error covariances for a large variety of different uncertainty sources. In the standard output files we do not provide the full averaging kernels (that would consider all the cross correlations between the different retrieval products) neither the full error covariances. The reason is that providing the full kernels and/or the full error covariances would strongly increase the storage needs for the data output (Weber, 2019).

Figure 8 explains the matrix blocks that are made available in the extended output file and in all standard output files. The extended file contains the full gain matrices, the Jacobian matrices for all state vector components, and Jacobians for parameters that are not retrieved but that affect the retrieval (spectroscopy, different cloud types, and surface emissivity). Using the gain matrices and the Jacobians, the full averaging kernels and the full error covariances can be calculated as indicated by Fig. 8. The full averaging kernel for the trace gas products is marked at the right side by the thick black frame (an example for these kernels is plotted in Fig. 9). The full error covariances are indicated by the yellow frame (examples of the root-mean-square values of the diagonals of these error covariances are plotted in Fig. 12).

The parts of this full matrix that are provided by the standard output files for all individual retrievals are indicated as the matrix blocks filled by green and red colour. Green represents the individual averaging kernels of the water vapour isotopologues, the greenhouse gases, $HNO_3$, and the atmospheric temperature. Red marks the cross kernels of the trace gas products with respect to atmospheric temperature (i.e., they inform how errors in the EUMETSAT L2 PPF atmospheric temperatures – used as MUSICA IASI a priori temperatures – affect the retrieved trace gas products). These temperature cross kernels allow the calculation of the full error covariances for the temperature uncertainty for each individual observation of the standard output file. In addition, the standard output files contain for all individual observatios square root values of the diagonal of the error covariance matrix for the most important uncertainty sources (noise and temperature uncertainty).

We provide differential or derivatives (covariances, averaging kernels, gain matrices and Jacobian matrices) related to the trace gas products always in the logarithmic scale. Logarithmic scale kernels are the same as the fractional kernels used in Keppens et al. (2015). Furthermore, we strongly recommend the use of the logarithmic scale kernels for analytic calculation. Because the MUSICA IASI trace gas retrievals are made on the logarithmic scale, the assumption of a moderately non-linear



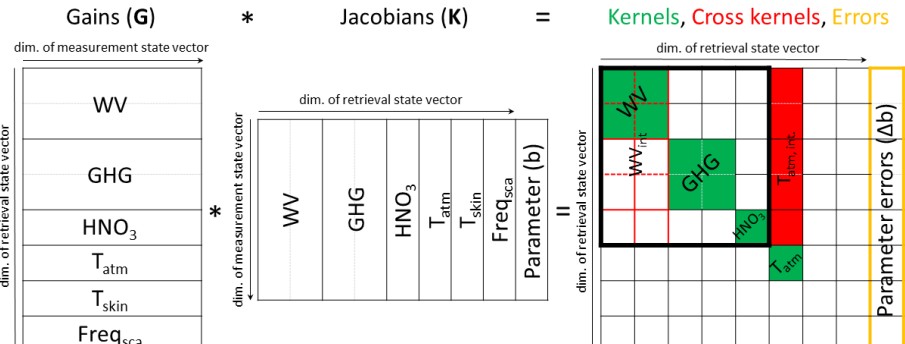

**Figure 8.** Schematic explanation of the Gain and Jacobian matrices provided in the extended netcdf output file, and the kernel and cross kernel matrices (indicated as the matrix blocks filled by green and red coulor, respectively) provided in all netcdf output files.

case according to Rodgers (2000) can be made on logarithmic scale (i.e. requires the use of logarithmic scale kernels), but has limited validity on the linear scale. More details on the valid assumption of a moderately non-linear problems are given in Appendix B.

### 5.2.1 Averaging kernels

Figure 9 depicts the averaging kernels for the full atmospheric composition state (water vapour proxy state, $N_2O$, $CH_4$, and $HNO_3$) for a typical summer time observation over a mid-latitudinal land location. Shown are all the matrix blocks marked by the thick black frame in the right part of the schematic of Fig, 8. In the diagonal we see the trace gas specific kernels and in the outer diagonal blocks the cross kernels. For the $H_2O$ proxy (see Sect. 4.4.2) we achieve very high values of about $5.3$ for DOFS (degree of freedom of signal, which is calculated as the trace of the respective matrix block). Also for the $\delta D$ proxy, $N_2O$, and

$CH_4$ the DOFS values are clearly larger than $1.0$, indicating the capability of the retrieval of providing some information of the trace gases' vertical distribution.

The cross kernel representing the impact of atmospheric $\delta D$ on the retrieved $H_2O$ ($\mathbf{A'_{12}}$ in Fig. 9) has the largest entries of all cross kernels; however, because variations in $\delta D$ are by an order of magnitude smaller than variations in $H_2O$, in reality this impact will be of secondary importance only. For consistency with the other data products we provide these kernels in the

$\{\ln[H_2O], \ln[HDO]\}$ basis (not in the $\{\frac{1}{2}(\ln[H_2O] + \ln[HDO]), (\ln[HDO] - \ln[H_2O])\}$ proxy basis used in Fig. 9). In the $\{\ln[H_2O], \ln[HDO]\}$ basis the cross kernels have very large and important entries and we provide in all standard files all four blocks of the water vapour isotopologue kernels (the diagonal kernels and the cross kernels).

Similarly we also provide in all standard files all four block kernels describing the greenhouse gases (kernels $\mathbf{A_{33}}$, $\mathbf{A_{34}}$, $\mathbf{A_{43}}$, and $\mathbf{A_{44}}$ in Fig. 9). Although the respective cross kernel values are rather small their availability supports the precise

characterisation of a combined $CH_4/N_2O$ product, which has a higher precision than the individual $N_2O$ and $CH_4$ products (see discussion in García et al., 2018).



**Figure 9.** Example of an averaging kernel for the full atmospheric composition state vector that is optimally estimated by the MUSICA IASI retrieval procedure: $\{\frac{1}{2}(\ln[H_2O] + \ln[HDO]), (\ln[HDO] - \ln[H_2O]), \ln[N_2O], \ln[CH_4], \ln[HNO_3]\}$. The kernel is for the 30. Aug. 2008 example observation over Lindenberg used for Fig. 4. Please note that $\frac{1}{2}(\ln[H_2O] + \ln[HDO])$ and $(\ln[HDO] - \ln[H_2O])$ are good proxies for $H_2O$ and $\delta D$, respectively.

Because $HNO_3$ has only weak spectroscopic signatures in the analysed spectral window, the respective kernel ($\mathbf{A_{55}}$ in Fig. 9) reveals a pronounced maximum, which is limited to the lower/middle stratosphere. By tuning the constraint (see discussion



**Table 2.** Metrics for sensitivity and resolution calculated from the elements of the averaging kernel matrix ($A_{i,j}$).

| Name | Formula | Represented characteristic and unit |
|---|---|---|
| Degree of Freedom for Signal (DOFS) | $\text{DOFS} = \sum_i A_{i,i}$ | Independent pieces of vertical information [unitless] |
| Measurement Response (ML) | $\text{MR}(i) = \sum_j A_{i,j}$ | Sensitivity [unitless] |
| Layer Width per DOFS (LWpD) | $\text{LWpD}(i) = \frac{\Delta z_i}{A_{i,i}}$ | Vertical resolution [km] |
| Centre (C) | $\text{C}(i) = \frac{\sum_j z_j A_{i,j}^2 \Delta z_j}{\sum_j A_{i,j}^2 \Delta z_j}$ | Vertical information displacement [km] |
| Resolving Length (RL) | $\text{RL}(i) = 12 \frac{\sum_j \left(z_j - \text{C}(i)\right)^2 A_{i,j}^2 \Delta z_j}{\left(\sum_j A_{i,j} \Delta z_j\right)^2}$ | Vertical resolution [km] |

for $1 < i < \text{nal}$: $\Delta z_i = \frac{z_{i+1} - z_i}{2} - \frac{z_i - z_{i-1}}{2}$; $\Delta z_1 = \frac{z_2 - z_1}{2} - z_1$; $\Delta z_{\text{nal}} = z_{\text{nal}} - \frac{z_{\text{nal}} - z_{\text{nal}-1}}{2}$

at the end of Sect. 4.6), we obtain DOFS values of generally close to 1.0. We also provide atmospheric temperature profile
kernels (not shown in Fig. 9), for which we typically obtain a DOFS value of about 2.0.

Because we want to provide averaging kernels for each individual observation, we developed a compression procedure,
which is necesary for keeping the size of the data files in an acceptable range. Section 5.2.4 describes the compression method,
the format, and the variables in which the averging kernels are provided.

### 5.2.2 Metrics for sensitivity and resolution

Table 2 gives an overview on metrics that can be calculated from the averaging kernel elements. In the previous section the
DOFS metric has been introduced as the trace of the averaging kernel matrix. Figure 10 depicts the typical geographical
distribution of the DOFS values for the different trace gas products. Largest values are generally achieved at low latitudes,
except for $HNO_3$, where we get largest values in middle and high latitudes. The high values for $H_2O$ indicate that we can
detect $H_2O$ profiles everywhere around the globe, but in particular in low latitudes. For $\delta$D and $CH_4$ we can also detect two
independent altitude layers in the tropics and summer hemispheric sub-tropics. There is limited profiling capability for $N_2O$
and almost no profiling capability for $HNO_3$. For the latter we ocassionally find DOFS values of below 0.8 over the tropics,
arid subtropical areas, and the central Antarctic. The DOFS values are provided in the variables with the suffix `_dofs`.

Figure 11 shows vertical profiles of the averaging kernel metric measurement response (MR), layer width per DOFS (LWpD),
information displacement (difference between centre altitude and nominal altitude), and resolving length (RL). The depicted
profiles are for the averaging kernels of Fig. 9.

The measurement response is the sum along the row of the averaging kernel matrix (Eriksson, 2000; Baron et al., 2002).
It is provided in the variables with the suffix `_response`. If a retrieval provides a smoothed version of the truth, without
systematically pushing results towards greater or smaller values, the sum of the elements over each row of the averaging kernel
should be unity. Any deviation of the row-sums from unity thus hints at an influence of the constraint that is beyond pure
smoothing (von Clarmann et al., 2020). Depending on the trace gas we observe different altitudes with MR values close to

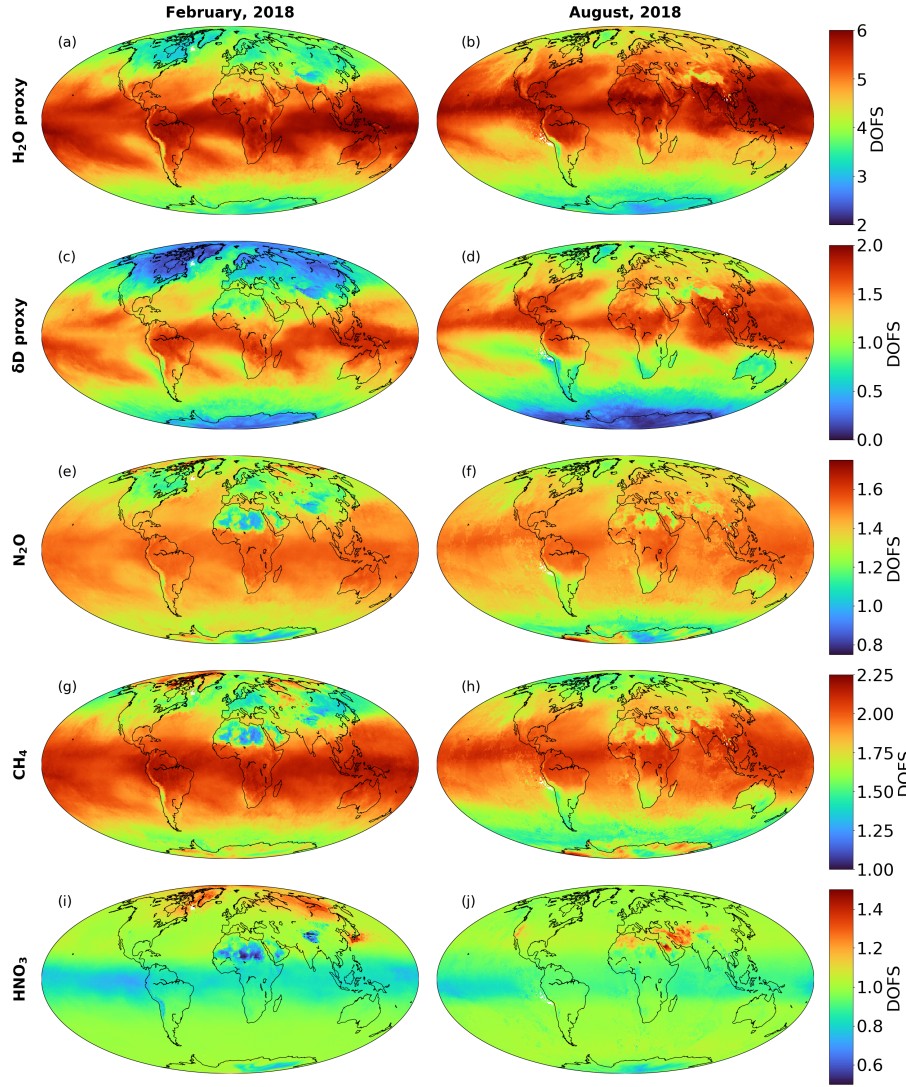

**Figure 10.** Monthly averaged maps of degree of freedom for signal (DOFS) values for February and August 2018 for all targeted atmospheric species: $H_2O$, $\delta D$, $N_2O$, $CH_4$, and $HNO_3$.

unity ($1 \pm 0.2$): tropospheric altitudes for $H_2O$ and $\delta D$, altitudes between the free troposphere and the lower stratosphere for $N_2O$ and $CH_4$, and lower stratospheric altitudes for $HNO_3$.

Layer width per DOFS is calculated as the local grid width divided by the respective diagonal value of the averaging kernel matrix (Purser and Huang, 1993; Keppens et al., 2015). It is a reasonable measure for vertical resolution. For our example observation we see a very good vertical resolution for $H_2O$ almost throughout the troposphere. For $\delta D$ the resolution is reasonable in the lower and middle troposphere, for $N_2O$ and $CH_4$ in the middle troposphere and upper troposphere/lower stratosphere, and for $HNO_3$ only in a very limited altitude region in the stratosphere. Maximum values in a row of the kernel

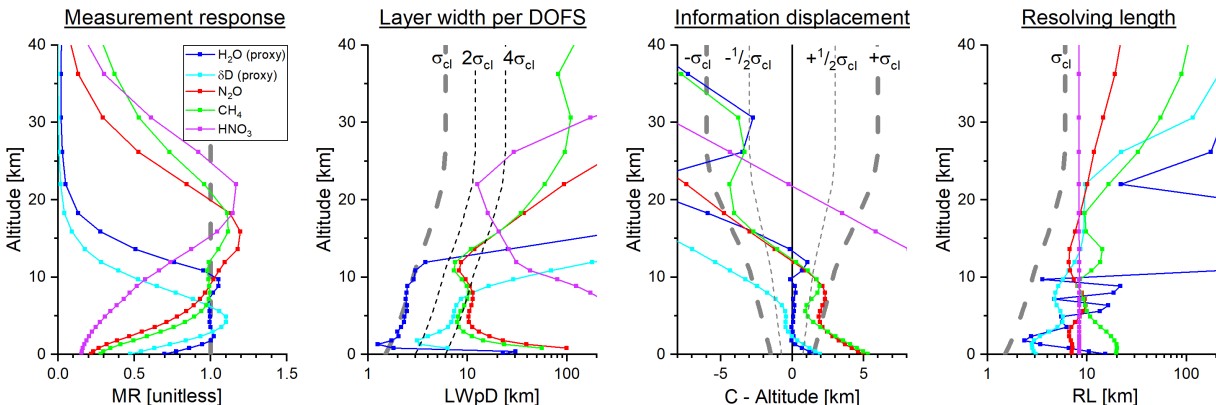

**Figure 11.** Profiles with averaging kernel metrics for the 30. Aug. 2008 example observation over Lindenberg used for Fig. 9. The thick grey dashed lines indicate for orientation the 100% value (for the panel showing the measurement response) and the apriori assumed vertical correlation length (for the panels showing the layer width per DOFS, information displacement, and resolving length).

matrix away from the diagonal means that the nominal altitude and the altitude of the maximum kernel values are different. For this altitudes LWpD values strongly increase, even if the MR value is still in an reasonable range (e.g. for $CH_4$ at about 500 15 km).

The centre altitude (C) informs about the atmospheric altitude region by which the retrieved values are mostly affected. In an optimal case this altitude region should correspond to the nominal altitude of the retrieval. Differences between the nominal altititude and the centre value reveal a vertical information displacement, i.e. the signals reported by the retrieval for the nominal altitude are real atmospheric signals of a systematically different altitude. We observe very low information 505 displacements for tropospheric $H_2O$ and middle tropospheric $\delta D$. For $N_2O$ and $CH_4$ the values are reasonable between the middle/upper troposphere and the lowermost stratosphere. For $HNO_3$ the centre value is almost the same for all altitudes, i.e. the signals retrieved at different altitudes reflect all the signals of the same real atmospheric altitude region.

The resolving length informs about the vertical resolution at the centre altitude, i.e. how broad is the atmospheric altitude layer by which the retrieved value is significantly affected. As briefly discussed in Rodgers (2000) resolving length is not a 510 satisfactory definition of resolution for slowly decaying averaging kernels or for averaging kernels that have strong sidelobes, for instance the MUSICA IASI kernels for $H_2O$ (see top left panel of Fig. 9).

Centre value and resolving length are calculated according to Eqs. (8) and (7) of Keppens et al. (2015). These parameters have been originally introduced by Backus and Gilbert (1970) and are also discussed in Chapter 3 of Rodgers (2000).

The variables with the suffix `_resolution` provide the vertical information displacement and resolution metrics for each 515 individual observation. As parameter 1 and 2 this variables provides the centre value (C) and the resolving length (RL), respectively, and as parameter 3 the layer width per DOFS value (LWpD).





**Table 3.** List of uncertainty assumptions used for the error estimations as shown in Fig. 12.

| Uncertainty source | Uncertainty assumption |
|---|---|
| Surface emissivity | 2%, with $100\,\text{cm}^{-1}$ spectral correlation length |
| Atmospheric temperature | Temperature uncertainty covariance according to `musica_apriori_cl` and `musica_at_apriori_amp` |
| Atmospheric water vapour (WV) | $H_2O$ and $\delta D$ proxy covariances ($\mathbf{S}_{\mathbf{a},H2O}$ and $\mathbf{S}_{\mathbf{a},\delta D}$ according to `musica_apriori_cl` and `musica_wvp_apriori_amp`) |
| WV continuum | +10% of model MT_CKD v2.5.2 |
| WV line intensity (coherent) | $\Delta \bar{S} = \Delta S_{H2O} = \Delta S_{HDO} = +5\%$ |
| WV pressure-broadening (coherent) | $\Delta \bar{\gamma} = \Delta \gamma_{H2O} = \Delta \gamma_{HDO} = +5\%$ |
| WV line intensity (incoherent) | $\Delta S' = -2\Delta S_{H2O} = 2\Delta S_{HDO} = +5\%$ |
| WV pressure-broadening (incoherent) | $\Delta \gamma' = -2\Delta \gamma_{H2O} = 2\Delta \gamma_{HDO} = +5\%$ |
| $N_2O$ line intensity | $\Delta S_{N2O} = +2\%$ |
| $N_2O$ pressure broadening | $\Delta \gamma_{N2O} = +2\%$ |
| $CH_4$ line intensity | $\Delta S_{CH4} = +2\%$ |
| $CH_4$ pressure broadening | $\Delta \gamma_{CH4} = +2\%$ |
| $HNO_3$ line intensity | $\Delta S_{HNO3} = +10\%$ |
| $HNO_3$ pressure broadening | $\Delta \gamma_{HNO3} = +10\%$ |
| Opaque cumulus cloud | 10% fractional cloud coverage, cloud top at 3 km |
| Mineral dust cloud | OPAC "Desert" (Table 4 in Hess et al., 1998), homogeneous, 2-4 km |
| Cirrus cloud | OPAC "Cirrus 3" (Table 1b in Hess et al., 1998), 25% fractional cloud coverage, 11-12 km |

### 5.2.3 Errors

For the 74 observations provided in the extended output file (see Sect. 3.1) calculations of a large variety of Jacobians and full gain matrices are available for a polar, mid-latitudinal, and tropical site (Borger et al., 2018). Figure 12 presents the errors
calculated for a mid-latitudinal summer observation using the gain matrices and Jocabians according to Eqs. (A10) and (A11). The uncertainty assumption $\Delta b$ and $\mathbf{S_b}$ used for these calculations are summarised in Table 3. The measurement noise error is calculated according to Eq. (A12) with $\mathbf{S_{y,noise}}$ being a diagonal matrix with diagonal values set to the mean square value calculated from the spectral residuals (measured - simulated spectra).

We organise the errors in three categories: random errors (measurement noise, uncertainties of emissivity and atmospheric
temperature, and interferences from atmospheric humidity and $\delta D$ variations), spectroscopic errors (uncertainties in the water continuum modelling and uncertainties in the intensity and pressure broading parameters of all target trace gases), and errors due to unrecognised clouds.

Concerning random errors, we find that atmospheric temperature uncertainties are dominating the error budget for all retrieval products except for $\delta D$ (because temperature uncertainties have similar impacts on $H_2O$ and HDO, they cancel out in



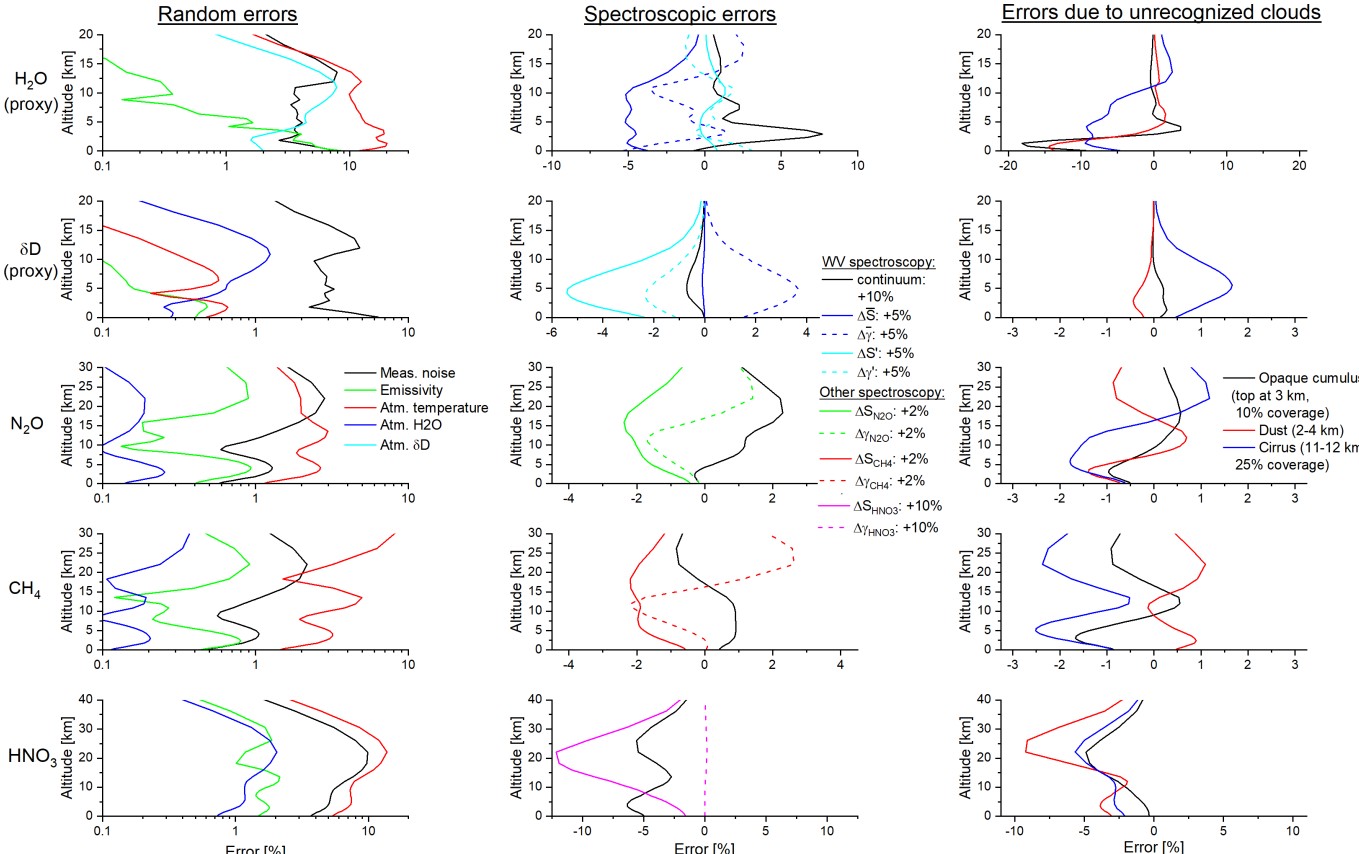

**Figure 12.** Example of error profiles estimated for all targeted atmospheric species: $H_2O$, $\delta D$, $N_2O$, $CH_4$, and $HNO_3$. Please note that for the species $H_2O$ and $\delta D$ the estimation is made for the respective proxies $\frac{1}{2}(\ln[H_2O] + \ln[HDO])$ and $(\ln[HDO] - \ln[H_2O])$. The errors are calculated according to the uncertainty assumptions of Table 3 and for the 30. Aug. 2008 example observation over Lindenberg used for Figs. 4 and 9.

their ratio). Measurement noise is the second most important error contributor (and the dominating error source for $\delta D$). Estimations of the dominating temperature error (assuming atmospheric temperature uncertainty covariances in line with August et al., 2012) and the measurement noise error are provided in standard files in the variables with suffix `_error`, for all trace gas products (for the water vapour isotopologue in the proxy state basis) and for atmospheric temperature.

By providing the cross averaging kernels with respect to atmospheric temperature (see matrix blocks filled by red colour at
the right side of the schematics of Fig. 8) we can calculate the propagation of any assumed temperature profile uncertainty $\Delta\boldsymbol{T}$ individually for all observations in the standard files, according to Eq. (A10):

$$\Delta\hat{\boldsymbol{x}} = -\mathbf{G}\mathbf{K_T}\Delta\boldsymbol{T} = -\mathbf{A_T}\Delta\boldsymbol{T}, \tag{15}$$



with $\mathbf{K_T}$ being the Jacobians for atmospheric temperature and $\mathbf{A_T}$ being the temperature cross kernel provided for all observations in the standard data file.

We can also reconstruct for all observations the full error covariance matrix $\mathbf{S_{\hat{x},noise}}$ due to the spectral noise used for constraining the solution state. For the MUSICA IASI processing we use as the spectral noise covariance $\mathbf{S_{y,noise}}$ a diagonal matrix with the mean-square values of the spectral residual (difference between simulated and measured spectrum). According to Eqs. (A5) to (A8) and (A12) we can write:

$$\mathbf{S_{\hat{x},noise}} = \mathbf{A(I-A)R}^{-1}. \tag{16}$$

$H_2O$ interferences from atmospheric $\delta D$, and $\delta D$ interferences from atmospheric $H_2O$ are also significant (blue and cyan lines in the random error plots of Fig. 12). For this reason we provide in the standard file the four blocks of the water vapour isotopologues averaging kernels, which enables us to estimate these interferences for each individual observation. The error covariance due to interference of $\delta D$ on $H_2O$ can be calculated by:

$$\mathbf{S_{\hat{x},\delta D,if}} = \mathbf{A'_{12}S_{a,\delta D}A'_{12}}^T, \tag{17}$$

and the error due to interference of $H_2O$ on $\delta D$ by:

$$\mathbf{S_{\hat{x},H2O,if}} = \mathbf{A'_{21}S_{a,H2O}A'_{21}}^T. \tag{18}$$

Here $\mathbf{S_{a,\delta D}}$ and $\mathbf{S_{a,H2O}}$ are covariances of the $\delta D$ and $H_2O$ proxy states, respectively, and $\mathbf{A'_{12}}$ and $\mathbf{A'_{12}}$ are the cross kernels of the proxy states. Please note that the water vapour isotopologue kernels provided in the standard files are for the $\{\ln[H_2O], \ln[HDO]\}$ basis and not for the $\{\frac{1}{2}(\ln[H_2O] + \ln[HDO]), (\ln[HDO] - \ln[H_2O])\}$ proxy state basis, i.e. for being used according to Eqs. (17) and (18) the provided kernels have to be transformed according to Eq. (4).

Spectroscopic uncertainties cause mainly systematic errors. The assumed uncertainty of line intensity $\Delta S$ and pressure broadening $\Delta \gamma$ (see Table 3) are in reasonable agreement with the values reported in Gordon et al. (2017). Respective error estimations can be performed for the 74 exemplary observations provided in the extended data file over a polar, mid-latitudinal and tropical site. As shown in Fig. 12 they are typically within 5%, except for $HNO_3$, where we estimate errors in the lower stratosphere due to spectroscopic uncertainties of up to 12% (mainly reflecting the larger uncertainty budget allowed for the band intensity). The uncertainties of the spectroscopic parameters of line intensity and pressure broadening mainly affect the retrieval of the trace gas, for which the parameters are assumed to be uncertain. Cross impacts are largest for uncertainties in water vapour parameters and there mostly for the water continuum (to a less extent for line intensity and pressure broadening). For this reason we plot the effect of the water continuum uncertainty for all trace gases, whereas the effects of the line intensity and pressure broadening parameters are only plotted for trace gas whose parameters are assumed to be uncertain.

MUSICA IASI retrievals are only executed when the EUMETSAT L2 PPF flag `flag_cldnes` is set to 1 (the IASI Instrumental Field Of View, IOFV, is clear) or 2 (the IASI IFOV is processed as cloud-free but small cloud contamination possible). This means that in particular for MUSICA IASI retrievals made with a cloud flag value of 2 clouds can have an impact, which should be examined. For this reason we calculated a variety of different cloud Jacobians for our 74 exemplary observations





over polar, mid-latitudinal and tropical sites and provide them in the extended data files. Examples for the obtained errors are depicted on the right of Fig. 12. We find that clouds with the properties as described in Table 3 have a significant effect on the retrievals. The impact of a cirrus cloud is in particularly strong and the $H_2O$ and $HNO_3$ data products seem to be the most affected. However, in this context we also have to consider the natural variability of the different trace gas products. Because the natural valiability of $\delta D$, $N_2O$, and $CH_4$ is very small, uncertainties due to clouds of 1% can already be a large problem. In

summary this estimation of errors due to unrecognized clouds indicates that we should be careful when using MUSICA IASI data products corresponding to an EUMETSAT L2 PPF cloud flag value of 2 (see also discussion in Sects. 6 and 7).

### 5.2.4 Matrix compression

In order to reduce the storage needs of the output files we compress the averaging kernel matrices. For this compression we perform a singular value decomposition of the original averaging kernel

$$\mathbf{A} = \mathbf{U}\mathbf{D}\mathbf{V}^T,$$ (19)

and a subsequent filtering for the leading eigenvalues. We only keep the most important eigenvalues and eigenvectors, i.e. we only keep a small part of the matrices $\mathbf{U}$, $\mathbf{D}$, and $\mathbf{V}$. The variables that store this leading information of the averaging kernels have in their names specific suffixes. The variable with suffix `_avk_rank` stores the number ($r$) of the leading eigenvalues and eigenvectors that are kept. Suffix `_avk_val` identifies the variable containing the eigenvalues. The variables with suffixes

`_avk_lvec` and `_avk_rvec` store the leading left and right eigenvectors. The reconstruction of the averaging kernel is made according to Eq. (19), whereby we setup the $r \times r$ diagonal matrix $\mathbf{D}$ consisting of the leading eigenvalues and the $n \times r$ matrices $\mathbf{U}$ and $\mathbf{V}$ consisting of the leading left and right eigenvectors. Here $n$ is the numbers of elements in the considered state vector. When reconstructing all four blocks of the water vapour or greenhouse gas averaging kernels, $n = 2nal$. For the reconstruction of the $HNO_3$ or atmospheric temperature averaging kernels $n = nal$. For more details on the effectiveness of

this compression method please refer to Weber (2019).

The suffixes `_xavkat_rank`, `_xavkat_val`, `_xavkat_lvec`, and `_xavkat_rvec` identify the respective variables needed for the reconstruction of the temperature cross averaging kernels. In this case the right eigenvectors have the length of the atmospheric temperature state vector, which is different from the length of the atmospheric state vector in case of the water vapour isotopologue and the greenhouse gas product (i.e. for the water vapour isotopologue and the greenhouse gas

temperature cross averaging kernels the left and right eigenvectors have different sizes).

### 6 Data filtering options

The MUSICA IASI retrieval data are provided with detailed information on the retrieval quality, the retrieval products' characteristics, and errors, as well as variables summarising cloud conditions and main aspects of sensitivity, vertical resolution and errors. In this section we discuss the variables providing these informations and recommend possibilities for data filtering.

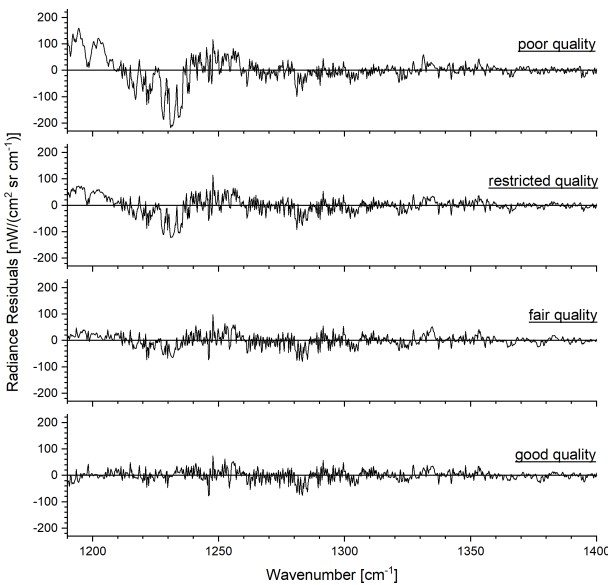

**Figure 13.** Example of residuals (simulated - measured spectra) belonging to the different quality groups as documented by the variable `musica_fit_quality_flag`: poor, restricted, fair, and good. The retrievals correspond to observations over Northern Africa with relatively high but very similar surface skin temperature (319.1-319.3 K) made during the Metop-A orbit #51267 as shown on the left of Fig. B1.

## 6.1 Clouds

The EUMETSAT L2 PPF flag `flag_cldnes` is written in the MUSICA IASI variable `eumetsat_cloud_summary_flag`. As discussed in Sect. 5.2.3 there is some risk that the MUSICA IASI product retrieved for `eumetsat_cloud_summary_flag` set to 2, has significant errors due to clouds. In order to exclude this risk we can filter out these data, i.e. we can use a very stringent cloud filtering criterion by only using observations where the variable `eumetsat_cloud_summary_flag` is set to '1'.

Another and less stringent option is to use in addition the EUMETSAT L2 fractional cloud cover, which is written in the MUSICA IASI variable `eumetsat_cloud_area_fraction`. If `eumetsat_cloud_summary_flag` is set to '2' we require in addition that the determination of a cloud area fraction has not been successful, i.e. we require that `eumetsat_cloud_area_fraction` is set to 'NaN'. No clear determination of a value for fractional cloud cover means that the cloud signals are rather weak (the contrast between cloud and surface signals are smaller than the instrument noise).

## 6.2 Quality of the spectral fit

The spectral noise level considered in the cost function (A2) during the MUSICA IASI processing is the root-mean-square (RMS) of the spectral fit residual (difference between simulated and measured spectrum). By this retrieval setting we consider as spectral noise level the degree to which the spectra can be understood by the forward model. The so defined spectral noise





level is generally larger than the pure instrumental noise level, because it is a sum of the instrumental noise and the signatures that are not understood by the foreward model. In the MUSICA IASI retrieval this RMS value is treated as white noise, i.e. for $\mathbf{S_{y,noise}}$ of the cost function (A2) we use a diagonal matrix filled by the mean-square values according to the spectral residuals.

As long as the residual is close to white noise, this kind of processing ensures a correct weighting of the measured spectra, on the one hand, and the a priori information, on the other hand. However, occasionally the measured spectra is very poorly
understood by the forward model and the residuals can not be described as a white noise instead the residuals show systematic signatures. This happens for instance, if incorrect surface emissivities are used or if the retrieval is made for an observation that is affected by a cloud. In order to identify the systematic part of the residuals we smooth the residuals using a $\pm 2\,\mathrm{cm}^{-1}$ running mean. The smoothed residuals are the systematic residuals and the difference of the original residuals and the smoothed residuals can then be interpreted as the random (or white noise) residuals. Residuals, systematic residuals, and random residuals
are provided in the standard files for each observation in the variable `musica_fit_quality`.

In order to facilitate the filtering of data corresponding to a poor spectral fit quality we set up a flag (provided as variable `musica_fit_quality_flag`) that works with the RMS values of the systematic residuals and the random residuals. The flag is set to 0 (poor quality) if the systematic residuals have an RMS value of larger than $40\,\mathrm{nW/(cm^2\,sr\,cm^{-1})}$. For all other observations we analyse the ratio between the RMS of the systematic residuals and the RMS of the random residuals. If this
ratio is larger than $1.0$ the flag is set to 1 (restricted quality), if it is between $0.5$ and $1.0$ the flag is set to 2 (fair quality), and if it is smaller or equal $0.5$ the flag is set to 3 (good quality). Figure 13 depicts residuals corresponding to different values of this fit quality flag. All observations are made during the same orbit, at closeby locations (Northern Africa), and for very similar surface temperatures. It is very likely that the poor spectral fit quality is due to incorrect surface emissivity values used for the respective retrievals (over arid areas like Northern Africa, surface emissivity data have an increased uncertainty, Seemann
et al., 2008). Our recommendation is to use data that belong to the quality groups fair and good.

### 6.3   Errors

The standard files provide for all obserations and all trace gas products estimations of the errors dominating the random error budget: errors due to noise in the spectra and errors due to uncertainties in the atmospheric temperature a priori data (the EUMETSAT L2 PPF temperatures). The noise error and estimations of atmospheric temperature error are given in the error
variable (variable with suffix `_error`, see Sect. 5.2.3) for all trace gas products and can be used for filtering out data with anomalously high errors.

Incorrect spectroscopic parameters (line intensity, pressure broadenig coefficients, or water continuum modelling) can be responsible for large errors. Although these uncertainty sources are systematic, the errors they cause depend on the sensitivity of the remote sensing system, which in turn is affected by the geometry of the observation. In first order the optical path of the
measured radiances depends on the platform zenith angle (PZA, provided as the variable `platform_zenith_angle`). In order to avoid that systematic uncertainties in the spectroscopic parameters cause artificial signals we can set threshold values for PZA and limit the PZA to angles close to nadir (e.g. by requiring PZA $\leq 30°$).





## 6.4 Sensitivity and resolution

The standard files provide the averaging kernels in a compressed format for all observations (see Sect. 5.2.4) as well as metrics
that capture the most important aspects of sensitivity and vertical resolution (see Sect. 5.2.2). These metrics are provided in the
variables with the suffixes `_response` and `_resolution` and allow analyses of sensitivity and vertical resolution for each
individual observation without the need for reconstructing the averaging kernels. We can use the metrics for filtering out data
where the response to the real atmospheric variability is low or where the vertical representativeness is irregular.

In order to ensure a good sensitivity (retrieval product being mainly affected by the real atmosphere and not by the a
priori assumption) the measurement response (MR) should be close to unity. Layer width per DOFS (LWpD), centre altitude
displacement (C − Altitude), and resolving length (RL) can be used to filter out data that do not fulfill the requirements in
terms of vertical representativeness needed for a dedicated study. Respective filter threshold values depend on the objective of
the scientific study. If processes within vertically well confined layers shall be exmined rather small vertical displacement and
very good vertical resolution is required and thus very stringent thresholds should be set.

Instead of filtering according to absolute values of LWpD, C−Altitude, or RL, the respective metrics allow the identification
of groups of data that have a similar vertical representativeness. For instance, we can robustly analyse time series of data that
have a stable vertical information displacement and a stable vertical resolution. For data where these conditions are not fulfilled,
time series signals might be significantly affected by the time variant data characteristics. The same is true when analysing
horizontal patterns, which might partly be due to the pattern in the data characetristics and not a real atmospheric pattern.

## 7   Data examples

Each Metop satellite accomplishes about 14 orbits per day, which makes about 5100 orbits per year. For our MUSICA IASI
retrieval period there are two or even three orbiting IASI instruments making operational measurements. Until the end of
October 2019 there were IASI-A and IASI-B and since November 2019 there is in addition IASI-C. So we have more than
10000 Metop/IASI orbits and in consequence MUSICA IASI netcdf output files per year with useful measurements (see
Sect. 3.1 for information on output data file nomenclature and format).

As an average about 30% of all measurements are made for cloud free conditions (EUMETSAT L2 PPF cloudiness assess-
ment summary flag set to '1' or '2', see also Sect. 4.1). This makes about 25000 individual retrievals per orbit/output file. In
the following we present examples of this large amount of data. We select example altitudes where the respective products
have generally a good sensitivity and reasonable vertical representativeness. According to Figs. 9 and 11 a good altitude choice
is 4.2 km for $H_2O$ and $\delta D$ and 10.9 km for $N_2O$ and $CH_4$. For $HNO_3$ the MUSICA IASI processor does not provide profile
information, instead the kernels for all altitudes show a similar vertical dependence and reveal retrieval sensitivity for a broad
lower stratospheric layer. For this reason we aggregate the $HNO_3$ data in form of partial column averaged mixing ratios for the
layer between 10 and 35 km. Details on this resampling are given in Appendix C.



**Table 4.** Filters applied for the time series and global daily maps of different species and for different altitudes (A) as shown in Figs. 14 and 15: EUMETSAT L2 PPF cloudiness assessment summary flag, MUSICA spectral fit quality flag, platform, zenith angle, MUSICA noise and temperature error, and MUSICA averaging kernel metrics (measurement response: $\mathrm{MR}$; sum of the partial column averaging kernel entries: $\sum_i A_i^*$; altitude layer width per DOFS: $\mathrm{LWpD}$; vertical information displacement: $\mathrm{C} - \mathrm{A}$).

| Species | Cloud | Fit quality | PZA | $\Delta x_{\mathrm{noise}}$ | $\Delta x_{\mathrm{Temp}}$ | $\lvert \mathrm{MR} - 1 \rvert$ | $\left\lvert \sum_i A_i^* - 1 \right\rvert^b$ | $\mathrm{LWpD}/\sigma_{\mathrm{cl}}{}^c$ | $\lvert \mathrm{C} - \mathrm{A} \rvert /\sigma_{\mathrm{cl}}$ |
|---|---|---|---|---|---|---|---|---|---|
| $H_2O$ (proxy) | 1, 2 | 2, 3 | – | $\leq 10\%$ | $\leq 30\%$ | $\leq 0.2$ | – | $[0.2, 2]$ | $\leq 0.5$ |
| $\delta D$ (proxy) | 1, 2 | 2, 3 | – | $\leq 25\%_o$ | $\leq 10\%_o$ | $\leq 0.2$ | – | $[0.3, 4]$ | $\leq 0.5$ |
| $N_2O$ | $1, 2^a$ | 2, 3 | $\leq 30°$ | $\leq 1.5\%$ | $\leq 5\%$ | $\leq 0.2$ | – | $[1.5, 5]$ | $\leq 1$ |
| $CH_4$ | $1, 2^a$ | 2, 3 | – | $\leq 1.5\%$ | $\leq 5\%$ | $\leq 0.2$ | – | $[1.5, 5]$ | $\leq 1$ |
| $HNO_3{}^b$ | 1, 2 | 2, 3 | – | $\leq 40\%^b$ | $\leq 40\%^b$ | – | $\leq 0.2^b$ | – | – |

[a]: only if variable `eumetsat_cloud_area_fraction` is set to 'NaN'

[b]: for dry air mixing ratios averaged for partial column 10-35 km a.s.l.

[c]: here the bottom thresholds are set to be below the lowest actually occurring positive value

## 7.1 Filtering

We filter the data according to the settings and threshold values of Table 4. For all data we requiring 'fair' and 'good' for the MUSICA IASI spectral fit quality (flag variable `musica_fit_quality_flag` is required to be set to '2' or '3') and we filter the data using the EUMETSAT L2 PPF cloudiness assessment flag (provided as the variable `eumetsat_cloud_summary_flag`). For the $N_2O$ and $CH_4$ data we apply a more stringent cloud filter and further inspect data where the EUMETSAT L2 PPF cloudiness assessment summary flag indicates a possibility of small cloud contamination. For respective data we require that

the EUMETSAT L2 processing cannot clearly attribute a value for fractional cloud cover, which means that the cloud signals are rather weak (see Sect. 6.1). We use this more stringent cloud filtering for $N_2O$ and $CH_4$, because both species have relatively weak atmospheric variabilities that are very similar to the errors estimated for a small cloud coverage (10% coverage with opaque cumulus clouds or 25% coverage with cirrus clouds).

Furthermore, we filter according to retrieval fit noise and estimated atmospheric temperature errors. The respective er-

rors are provided for each observation on the MUSICA IASI standard file output variable with suffix `_error`. For $HNO_3$ we calculate the retrieval fit noise and the estimated temperature errors for the 10-35 km partial column averaged mixing ratios according to Eq. (C7), whereby we reconstruct the noise covariance matrix for $HNO_3$ according to Eq. (16) and generate the atmospheric temperature covariance according to Eq. (7) using the MUSICA IASI standard file output variables `musica_at_apriori_amp` and `musica_apriori_cl` for setting up the values of $v_{\mathrm{amp},i}$ and $\sigma_{\mathrm{cl},i}$, respectively.

In order to ensure that the time series signals or horizontal patterns are not significantly affected by varying sensitivity and vertical resolution we filter $H_2O$, $\delta D$, $N_2O$, an $CH_4$ data according to the averaging kernel metrics $\mathrm{MR}$, $\mathrm{LWpD}$, and $\mathrm{C} - \mathrm{Altitude}$. This filters out data with anomalous vertical sensitivities. For $HNO_3$ we calculate the 10-35 km partial column

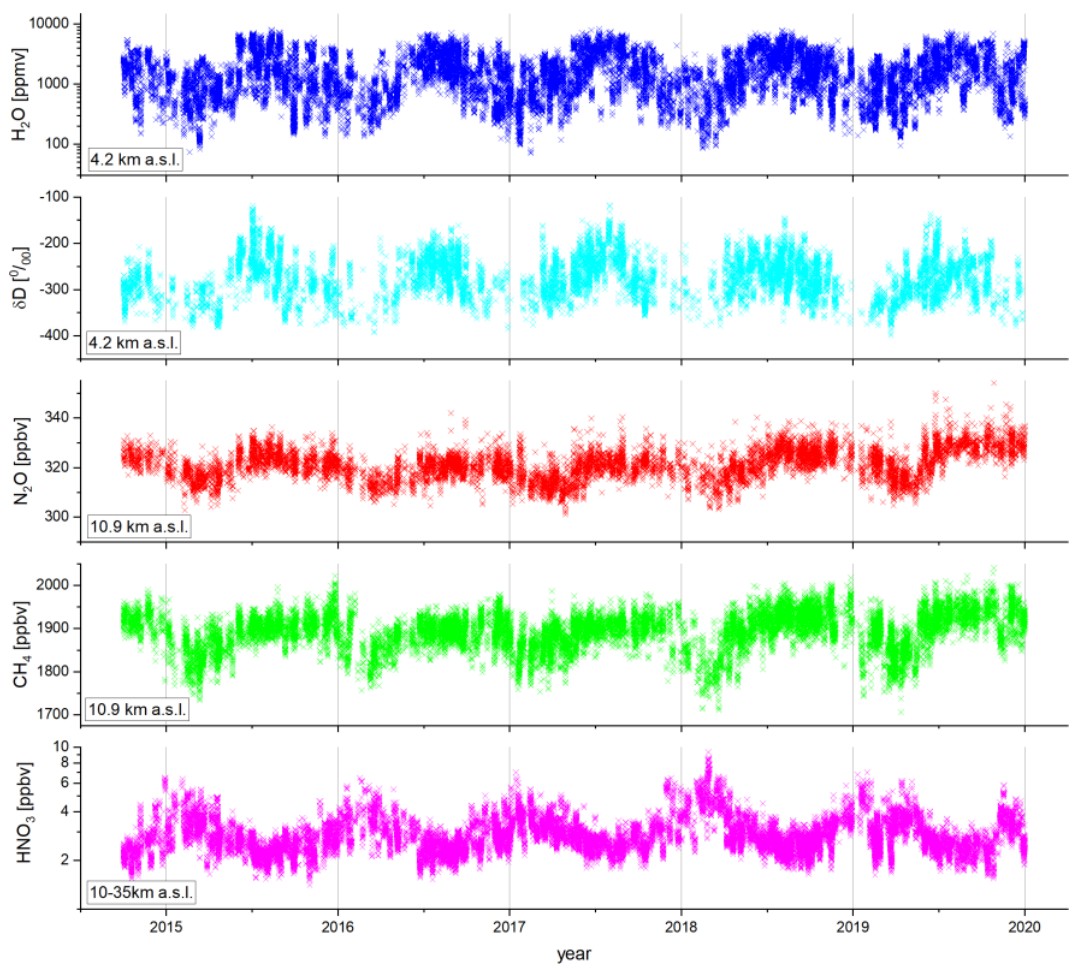

**Figure 14.** Time series of MUSICA IASI trace gas products at selected altitudes above Karlsruhe (48-49°N; 8-9°E): 4.2 km a.s.l. for $H_2O$ and $\delta D$, 10.9 km a.s.l. for $N_2O$ and $CH_4$, and 10-35 km a.s.l. for $HNO_3$. Data have been filtered according to the setting given in Table 4.

averaged mixing ratio averaging kernels according to Eq. (C6) and filter for good sensitivity by requiring a diagonal entry being close to unity.

The filter threshold values of $LWpD$ and $C - Altitude$ are defined relative to the a priori assumed vertical correlation length (provided in the variable `musica_apriori_cl`). By using this ratio we can work with the same threshold value for all altitudes, because as seen in Fig. 11, $LWpD$ and $C - Altitude$ have a similar vertical dependency as the vertical correlation length.

### 7.2 Continuous time series

In this section we give an example of the temporal continuity of the data. Figure 14 depicts a time series at the midlatitudinal site of Karlsruhe, Germany, between October 2014 and January 2020 of MUSICA IASI trace gas retrieval products. For all





trace gases, except for $\delta D$, we have a good temporal coverage with no significant data gaps caused by the comprehensive data filtering. Concerning $\delta D$, there is a reduced data amount in winter mainly due to filtering out data with reduced sensitivity (measurement response below 0.8). It is worth noting that for the $\{H_2O,\delta D\}$ pair optimal estimation product – generated a
posteriori according to Diekmann et al. (2021) – we a achieve a significantly better measurement response.

    We observe typical seasonal cycles for all species. The seasonal cycles of $H_2O$ and $\delta D$ follow the seasonal cycle of temperature. In winter $H_2O$ concentrations can be as low as 100 ppm and $\delta D$ values can be below $-350‰$. In summer the maximum values are about 8000 ppm and $-150‰$. The concentrations of $N_2O$ and $CH_4$ at 10.9 km a.s.l. are lowest in winter/spring and highest in summer/autumn. This cycle is linked to the vertical shift of the tropopause altitude: in winter/spring the 10.9 km
altitude is much stronger affected by the stratosphere (where $N_2O$ and $CH_4$ are decresing with height) than in summer/autumn. Concerning $HNO_3$ we observe highest values in winter/spring which might indicate the detection of air masses with an Arctic stratospheric history (Arctic winter stratospheric $HNO_3$ mixing ratios are particularly large).

### 7.3 Daily global maps

In this section we give an example of the good daily global coverage achieved by high quality MUSICA IASI products. Figure
15 depicts the data retained during 24 h when using the filter setting listed in Table 4. For our example we choose 1 February and 1 August 2018 and plot the data for the same altitudes as in Fig. 14. For all data products, except for $\delta D$, we have a very dense global coverage. Areas with missing data are mostly linked to the cloud filtering. The reduced data coverage for $\delta D$ in the mid and high latitudinal winter hemispheres is due to $\delta D$ measurement response values lying below 0.8 (we achive a significantly better measurement response and thus horizontal coverage for the optimal estimation $\{H_2O,\delta D\}$ pair product
generated according to Diekmann et al., 2021).

    Highest $H_2O$ concentrations at 4.2 km are observed at low latitudes where temperatures are generally highest. However, there are also low latitudinal areas with rather low $H_2O$ concentrations, for instance in the eastern Pacific on 1 August 2018, which indicates to a region where large scale subsidence is prevailing. The $\delta D$ values at 4.2 km are also highest at low latitudes; however with a stronger zonal variability. For high tropical $H_2O$ concentrations, $\delta D$ values can be relatively high (for instance
on 1. Feb. 2018 in the tropical Atlantic) or relatively low (for instance on 1 February 2018 in the tropical Indian Ocean). This indicates that $\delta D$ data contains information that is complementary to the $H_2O$ data.

    For $N_2O$ and $CH_4$ at 10.9 km we observe maximum concentrations at low latitudes and rather low values in the polar regions. The reason is that at high latitudes the 10.9 km altitude is strongly influenced by low stratospheric concentrations, whereas in the tropics the 10.9 km altitude is representative for the upper troposphere, where concentrations are higher. This means that
the concentrations observed at 10.9 km reflect to a large extent the altitude of the tropopause. On 1 August 2018 we observe for both trace gases a clear gradient between the northern and southern hemispheres, whereas there is no significant gradient on 1 February 2018. This is caused by higher tropospheric concentrations of both trace gases in the northern hemisphere. On 1 August 2018 the stratosphere affects the 10.9 km altitude stronger in the southern hemisphere than in the northern hemisphere and we observe a particular strong gradients. On 1 February 2018 it is the other way round and the tropospheric concentration
gradients are counterbalanced by the tropopause altitude effect.

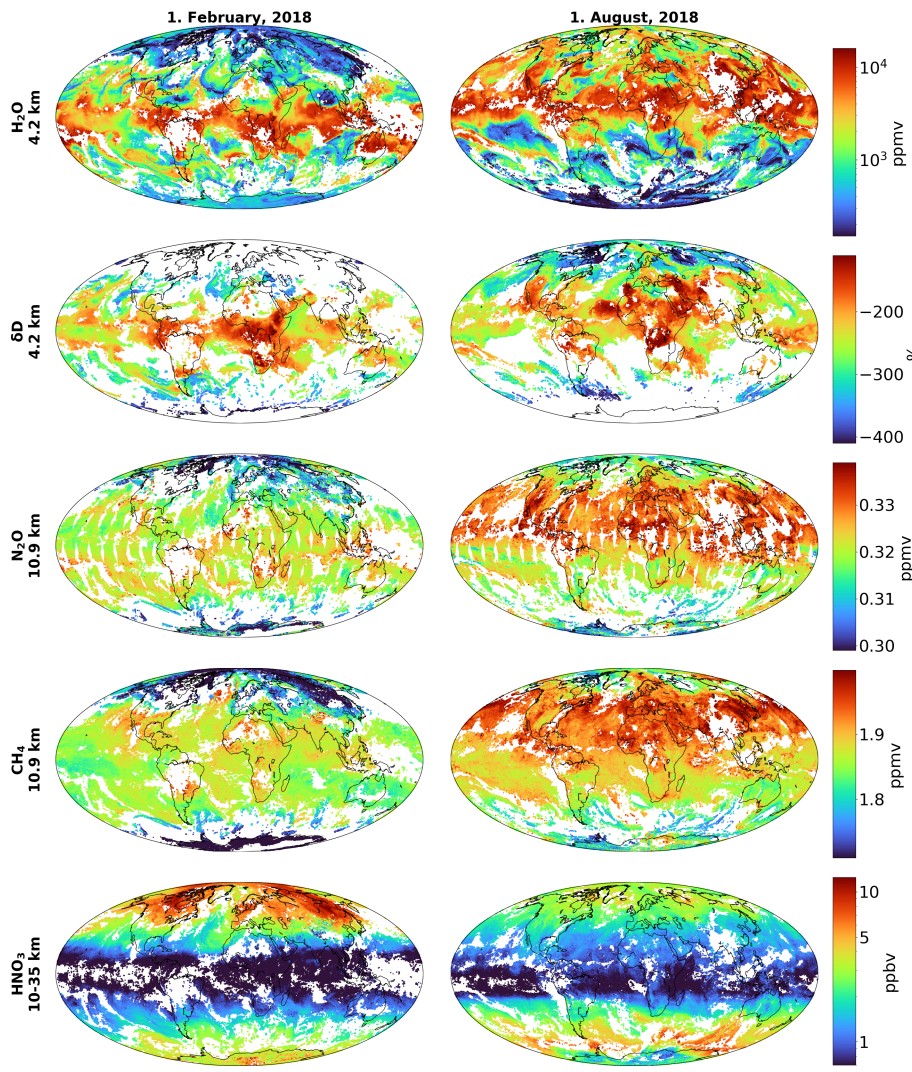

**Figure 15.** Daily maps for 1 February and 1 August 2018 of the trace gas products for the same altitudes as in Fig. 14. Altitude 4.2 km a.s.l. for $H_2O$ and $\delta D$, 10.9 km a.s.l. for $N_2O$ and $CH_4$, and 10-35 km a.s.l. for $HNO_3$. Data have been filtered according to the filter settings given in Table 4 (same as for Fig. 14).

The global maps of the $HNO_3$ 10-35 km partial column averaged mixing ratios show very low values in the tropics and highest values in polar regions. However, in Antarctic winter low values are also found, because at very low temperatures ($<$ 195 K) polar stratospheric clouds (PSCs) are formed on which $HNO_3$ condensates. In Arctic winter temperatures are generally not that low and PSCs and consequently low $HNO_3$ are mainly related to locally restricted mountain lee waves.



## 8 Interoperability and data reusage

The MUSICA IASI full retrieval product provides for each individual observation detailed information on retrieval settings (a priori and constraints) and retrieval characteristics (error covariances and averaging kernels). This comprehensive set of information ensures ultimate interoperability and offers possibility for a variety of data reuse applications, in particular, because the MUSICA IASI inversion problem is a moderately non-linear problem (see Appendix B). In the following we briefly list some data reusage possiblities.

For interoperability (the common use of different data sets or their inter-comparison) the impact of different a priori data should be assessed or eliminated. Assuming that the MUSICA IASI data (generated using a priori state $x_a$) should be commonly used with (or inter-compared to) another remote sensing data set whose retrieval processor used the a priori state $x_{a,m}$. Then we can calculate the MUSICA IASI retrieval state that would result from an $x_{a,m}$ a priori usage according to Eq. (B1). For these calculations we need from the side of the MUISCA IASI data, the originally retrieved state, the a priori state, and the averaging kernels, which is all provided by the MUSICA IASI full retrieval product.

For comparisons to atmospheric model simulation or for data assimilation applications a remote sensing product has to be made available together with full information about its error covariances and measurement operator. This is the case for the MUSICA IASI full retrieval product data set. For each individual observation the averaging kernels are made available and the full a posteriori covariances and the error covariances due to the fit residuals can be reconstructed from the provided constraint and the averaging kernel matrices according to Eqs. (A7) and (16), respectively.

As shown in Sect. 7 and Appendix C the MUSICA IASI trace gas profiles can be easily resampled according to user specific needs in form of partial column averaged mixing ratios with corresponding averaging kernels and error covariances. This is possible, because the data set provides the full information on pressure profiles, constraints (for reconstructing the error covariances due to the corresonding fit residuals, see Eq. (16)), temperature cross kernels $\mathbf{A_T}$ (in order to calculate the error covariances due to atmospheric temperature uncertainties, see Eq. (15)), and averaging kernels.

Worden et al. (2012) and García et al. (2018) discussed the advantages of a $CH_4/N_2O$ ratio product. García et al. (2018) showed that this ratio product has a theoretically higher precision than the individual $N_2O$ and $CH_4$ products. Because $N_2O$ is in the troposphere chemically more stable than $CH_4$ it is also more homogeneously distributed than $CH_4$. García et al. (2018) argued that by combining $CH_4/N_2O$ ratio observations with a model of the $N_2O$ climatology, it should be able to determine tropospheric $CH_4$ concentration with relatively high precision. The MUSICA IASI full retrieval product prodvides informations on constraints and the averaging kernels (including the cross averaging kernels between $N_2O$ and $CH_4$) thus it offers all needed for calculating the $CH_4/N_2O$ ratio product as well as the corresponding averaging kernels and error covariances.

Another interesting data reuse possibility is that the retrievals' a priori data or the retrievals' constraints can be modified a posteriori in accordance to particular user requirements. According to Eq. (18) of Rodgers and Connor (2003) we can calculate the retrieval result ($\hat{x}_m$) for a modified constraint ($\mathbf{R_m}$) by:

$$\hat{x}_m = x_a + \mathbf{R_m}^{-1}\mathbf{A}^T(\mathbf{A}\mathbf{R_m}^{-1}\mathbf{A}^T + \mathbf{S}_{\hat{x},\text{noise}})^{-1}(\hat{x} - x_a). \tag{20}$$





Here $x_a$, $\mathbf{A}$, $\mathbf{S_{\hat{x},noise}}$, and $\hat{x}$ are the a priori state, the averaging kernel, the error covariance due to retrieval fit noise, and the originally retrieved state, respectively. All this information is made available in (or can be reconstructed from the information

provided by) the MUSICA IASI full retrieval product. Diekmann et al. (2021) presents an an optimal estimation {$H_2O$,$\delta D$} pair product, which among others makes use of such a posteriori constraint modification.

Schneider et al. (2021c) presents another possibility for MUSICA IASI data reuse. They apply the extensive information provided in the MUSICA IASI full retrieval product for optimally combining MUSICA IASI $CH_4$ data with the total column $XCH_4$ retrieval products of the sensor TROPOMI (TROPOspheric Monitoring Instrument) aboard the satellite Sentinel-5P

(Lorente et al., 2020) without the need for running new retrievals. This a posteriori product combination can be achieved by Kalman filter calculations (Kalman, 1960; Rodgers, 2000), which have large similarities to Eq. (20). The method optimally combines the MUSICA IASI retrieval state (vector $\hat{x}$) with the information provided by the TROPOMI $XCH_4$ product (the scalar $\hat{x}_n$, we use index $n$ for new observation):

$$\hat{x}_c = \hat{x} + \hat{\mathbf{S}}a_n(a_n{}^T \hat{\mathbf{S}} a_n + S_{\hat{x}_n,noise})^{-1}[\hat{x}_n - x_a - (a_n{}^T \hat{x} - a_n{}^T x_a)]. \tag{21}$$

Here the vector $\hat{x}_c$ is the optimally combined state, the row vector $a_n{}^T$ is the column averaging kernel of the TROPOMI $XCH_4$ observation, the scalar $x_a$ is the a priori $XCH_4$ data, and the vector $x_a$ the a priori $CH_4$ profile. $\hat{\mathbf{S}}$ is a posteriori covariance of the MUSICA IASI data, which can be reconstructed with averaging kernel and constraint matrices being available according to Eq. (A7). The scalar $S_{\hat{x}_n,noise}$ is the measurement noise error variance of the TROPOMI $XCH_4$ product. Optimal means here that the uncertainties and sensitivities of the MUSICA IASI $CH_4$ product and the TROPOMI $XCH_4$ product are correctly

taken into account.

## 9  Summary and outlook

Measurements of the IASI instruments on the three satellites Metop-A, -B, and -C have been processed by the MUSICA IASI processor. The processing has been made globally for all measurements that are declared as likely cloud free by the EUMETSAT L2 PPF cloud detection procedure. Here we report on the full retrieval product of the MUSICA IASI processing

version 3.2.1 for the time period between October 2014 and June 2019.

The full retrieval product is the comprehensive output of the main MUSICA IASI processing chain. It contains the simulated and the residual radiances (the difference between measured and simulated radiances), some flags and retrieval outputs provided by the EUMETSAT L2 PPF processing, full information on the MUSICA IASI retrieval settings and the full MUSICA IASI retrieval output. For each observation we provide information on the MUSICA IASI a priroi settings and constraints, so that the

data are very easily reproduceable. The retrieval output are the trace gas profiles of $H_2O$, $HDO$, $N_2O$, $CH_4$, and $HNO_3$ as well as the atmospheric temperature profiles. Concerning $H_2O$ and $HDO$ the retrieval is optimised for $H_2O$ and the ratio of $HDO/H_2O$. All products are provided with a very extensive characterisation. For each individual retrieval the leading errors are made available together with the averaging kernels. In order to reduce the data volume the kernels are provided in a compressed data format and can be reconstructed by simple matrix calculations. In addition we provide variables with averaging kernels metrics

that capture the most important characteristics of the vertical representativeness (sensitivity and vertical resolution). These



variables can be used for identifying data with an acceptable vertical representativeness without the need for reconstructing the averaging kernels. We give some suggestions on how to use different flags, error information, and averaging kernel metrics for data filtering recommendable for the study of global distribution maps or time series.

The output of a priori states and averaging kernels for each individual observation guarantees ultimate interoperability (the
common use of different data sets or their inter-comparison). Furthermore, the additinal supply of constraint matrices for each individual observation together with the averaging kernels enables us to reconstruct the a posteriori covariances and the retrieval fit noise error covariance. Having all these information available offers excellent data reuse possibilities. We can a posteriori adjust the a priori or the constraints to specific user needs or optimally combine the MUSICA IASI products with other remote sensing products without the need for running new retrievals.

Currently MUSICA IASI data are available until June 2019. Nevertheless, the processing is ongoing. For IASI observations starting after June 2019 the MUSICA IASI processing version 3.3.0 instead of 3.2.1 is used. In version 3.2.1 there are some very minor inconsistencies in setting up the vertical gridding and in setting the a priori of $\delta D$ and the constraints for $N_2O$, $CH_4$, and $HNO_3$, which are accounted for during the postprocessing step. In version 3.3.0 these inconsistencies are already addressed before running the retrievals. This is the only difference between the two processing versions and it is actually not
noticeable by the data user. The here provided report on version 3.2.1 data is equally valid for version 3.3.0 data. MUSICA IASI data for observations in the second half of 2019 and in 2020 (processed using version 3.3.0) will soon be made available for the public in the same format as the here presented data.

## 10 Data availability

The MUSICA IASI data can be freely downloaded at http://www.imk-asf.kit.edu/english/musica-data.php. We offer two data
packages with DOI. The first data package has a data volume of about 17.5 GB and is linked to https://doi.org/10.35097/408 (Schneider et al., 2021b). It contains example standard output data files for all MUSICA IASI retrievals made for a single day (more than 0.6 million) and a description of how to access the total dataset (2014-2019, data volume 25 TB) or parts of it. This data package is for users interested in the typical global daily data coverage and in information about how to download the large data volumes of global daily data for longer periods. The second data package contains the extended output data file, has
only about 73 MB, and is linked to https://doi.org/10.35097/412 (Schneider et al., 2021a). It contains retrieval products for only 74 observations made at a polar, mid-latitudinal and tropical location. It provides the same variables as the standard output files and in addition the variables with the prefixes `musica_jac_` and `musica_gain_`, which are Jacobians for many different uncertainty sources and Gain matrices (due to this additional variables it is called the extended output file). Because this data package is rather small, it is recommended to potential reviewers and to users for having a quick look on the data.





## Appendix A: Basics of retrieval theory and notations

This appendix gives an overview on the thereotical basics and notations of optimal estimation remote sensing retrieval methods. It is meant as a compilation of the most important equations that are related to the discussions provided in this paper. Although it is similar to Section 2.1 of Borger et al. (2018), we think it is here a very helpful support for readers that are no experts in the field. Further details on remote sensing retrievals can be found in Rodgers (2000). For a general introduction on vector and matrix algebra we recommend dedicated textbooks.

Atmospheric remote sensing means that the atmospheric state is retrieved from the radiation measured after having interacted with the atmosphere. This interaction of radiation with the atmosphere is modeled by a radiative transfer model (also called the forward model, $\boldsymbol{F}$), which enables relating the measurement vector and the atmospheric state vector by:

$$\boldsymbol{y} = \boldsymbol{F}(\boldsymbol{x}, \boldsymbol{b}). \tag{A1}$$

We measure $\boldsymbol{y}$ (the measurement vector, e.g. a thermal nadir spectrum in the case of IASI) and are interested in $\boldsymbol{x}$ (the atmospheric state vector). Vector $\boldsymbol{b}$ represents auxiliary parameters (like surface emissivity) or instrumental characteristics (like the instrumental line shape), which are not part of the retrieval state vector. However, a direct inversion of Eq. (A1) is generally not possible, because there are many atmospheric states $\boldsymbol{x}$ that can explain one and the same measurement $\boldsymbol{y}$.

For solving this ill-posed problem a cost function $J$ is set up, that combines the information provided by the measurement with a priori known characteristics of the atmospheric state:

$$J = [\boldsymbol{y} - \boldsymbol{F}(\boldsymbol{x}, \boldsymbol{b})]^T \mathbf{S}_{\mathbf{y},\mathbf{noise}}^{-1} [\boldsymbol{y} - \boldsymbol{F}(\boldsymbol{x}, \boldsymbol{b})] + [\boldsymbol{x} - \boldsymbol{x}_a]^T \mathbf{R} [\boldsymbol{x} - \boldsymbol{x}_a]. \tag{A2}$$

Here, the first term is a measure of the difference between the measured spectrum (represented by $\boldsymbol{y}$) and the spectrum simulated for a given atmospheric state (represented by $\boldsymbol{x}$), while taking into account the actual measurement noise ($\mathbf{S}_{\mathbf{y},\mathbf{noise}}$ is the measurement noise covariance matrix). The second term of the cost function (A2) constrains the atmospheric solution state ($\boldsymbol{x}$) towards an a priori most likely state ($\boldsymbol{x}_a$), whereby the kind and strength of the constraint are defined by the constraint matrix $\mathbf{R}$, for which we use an approximate inversion of the a priori covariance matrix $\mathbf{S_a}$ (more details see Sect. 4.6):

$$\mathbf{R} \approx \mathbf{S_a}^{-1}. \tag{A3}$$

The constrained solution is reached at the minimum of the cost function (A2). Due to the non-linear behavior of $\boldsymbol{F}(\boldsymbol{x}, \boldsymbol{b})$, the minimisation is generally achieved iteratively. For the $(i+1)$th iteration it is:

$$\boldsymbol{x_{i+1}} = \boldsymbol{x_a} + \mathbf{G_i}[\boldsymbol{y} - \boldsymbol{F}(\boldsymbol{x_i}, \boldsymbol{b}) + \mathbf{K_i}(\boldsymbol{x_i} - \boldsymbol{x_a})]. \tag{A4}$$

$\mathbf{K}$ is the Jacobian matrix (derivatives that capture how the measurement vector will change for changes in the atmospheric state $\boldsymbol{x}$). $\mathbf{G}$ is the gain matrix (derivatives that capture how the retrieved state vector will change for changes in the measurement vector $\boldsymbol{y}$). $\mathbf{G}$ can be calculated from $\mathbf{K}$, $\mathbf{S}_{\mathbf{y},\mathbf{noise}}$ and $\mathbf{R}$ as:

$$\mathbf{G} = (\mathbf{K}^T \mathbf{S}_{\mathbf{y},\mathbf{noise}}^{-1} \mathbf{K} + \mathbf{R})^{-1} \mathbf{K}^T \mathbf{S}_{\mathbf{y},\mathbf{noise}}^{-1}, \tag{A5}$$





with the a posteriori covariance matrix ($\hat{\mathbf{S}}$):

$$\hat{\mathbf{S}} = (\mathbf{K}^T \mathbf{S_{y,noise}}^{-1} \mathbf{K} + \mathbf{R})^{-1}, \qquad (A6)$$

which can also be written as:

$$\hat{\mathbf{S}} = (\mathbf{I} - \mathbf{A})\mathbf{R}^{-1}, \qquad (A7)$$

where $\mathbf{I}$ is the identidy operator and $\mathbf{A}$ the averaging kernel matrix.

The averaging kernel is an important component of a remote sensing retrieval and it is calculated as:

$$\mathbf{A} = \mathbf{GK}. \qquad (A8)$$

The averaging kernel $\mathbf{A}$ reveals how a small change of the real atmospheric state vector $\boldsymbol{x}$ affects the retrieved atmospheric state vector $\hat{\boldsymbol{x}}$:

$$\hat{\boldsymbol{x}} - \boldsymbol{x}_a = \mathbf{A}(\boldsymbol{x} - \boldsymbol{x}_a). \qquad (A9)$$

The propagation of errors due to parameter uncertainties $\Delta b$ can be estimated analytically with the help of the parameter Jacobian matrix $\mathbf{K_b}$ (derivatives that capture how the measurement vector will change for changes in the parameter $\boldsymbol{b}$). According to Eq. (A4), using the parameter $\boldsymbol{b} + \Delta b$ (instead of the correct parameter $\boldsymbol{b}$) for the forward model calculations will result in an error in the atmospheric state vector of:

$$\boldsymbol{\Delta}\hat{\boldsymbol{x}} = -\mathbf{GK_b}\Delta b. \qquad (A10)$$

The respective error covariance matrix $\mathbf{S_{\hat{x},b}}$ is:

$$\mathbf{S_{\hat{x},b}} = \mathbf{GK_b S_b K_b}^T \mathbf{G}^T, \qquad (A11)$$

where $\mathbf{S_b}$ is the covariance matrix of the uncertainties $\Delta b$.

Noise on the measured radiances also affects the retrievals. The error covariance matrix for noise can be analytically calculated as:

$$\mathbf{S_{\hat{x},noise}} = \mathbf{GS_{y,noise} G}^T, \qquad (A12)$$

where $\mathbf{S_{y,noise}}$ is the covariance matrix for noise on the measured radiances $\boldsymbol{y}$.

Note that Eqs. (A5) to (A12) are only valid for a moderately non-linear inversion problem (see Chapter 5 of Rodgers, 2000). In Appendix B we show that our inversion problem is of such kind.

## Appendix B: Linearity

As outlined in Sect. 4 the MUSICA IASI processor uses a logarithmic scale for constraining the trace gas retrievals. We strongly recommend to work on the logarithmic scale for the analytic treatment of the trace gas states. This is very obvious



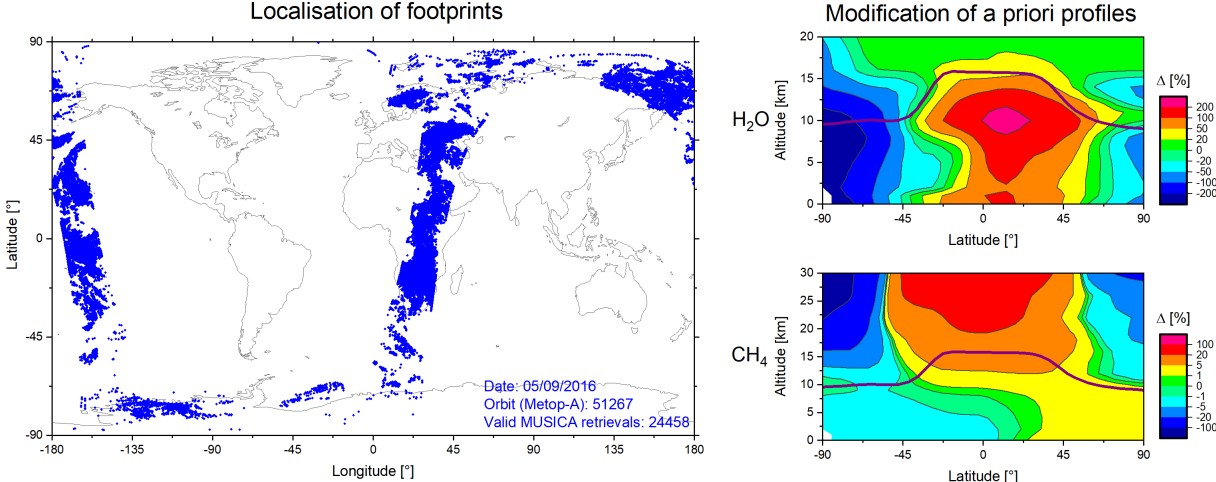

**Figure B1.** Setup of the linearity test using modified a priori data. The left panel shows the localisation of the footprints of the used examplary orbit and the right panels depicts latitudinal cross sections documenting the modification of the $H_2O$ and $CH_4$ a priori data (modified - original, thick solid violett line indicates the tropopause altitude).

in the context of the water vapour isotopologue proxy introduced in Sect. 4.4.2 (a transformation to the proxy state is only possible on the logarithmic scale). In addition, the analytic treatment of the states is important for characterising the data in the context of Eqs. (A9)-(A12) or Eqs. (15)-(18) and it can also be used for modifiying the retrieval settings without the need of

900 performing new computationally expensive retrieval calculations (see Chapter 10 of Rodgers, 2000). However, a requirement for the analytic treatment is that the problem is moderately non-linear (linearisation is adequate for the analytic treatment not for finding the solution, see Chapter 5 of Rodgers, 2000). In this Appendix we demonstrate that our problem is indeed moderately non-linear as long as we perform the calculations on the logarithmic scale.

## B1 Setup of the linearity test

We test the validity of assuming linearity for the analytic treatment by performing retrievals with different a priori settings. The standard setting is described in Sect. 4.5. It has a dependence on latitude as well as on seasonal and interanual time scales. For the test we perform additional retrievals with a priori data that have no latitudinal dependence, i.e. we use for all latitudes a latitudinal mean a priori profile. The additional retrievals are made for the Metop-A orbit #51267, whose footprints are depicted on the left of Fig. B1. We choose this orbit because it has a good global representativeness: the first part consists of observations

over land and covers many different latitudes (Western Asia to South Africa) and the second part of observation over sea from pole-to-pole (Pacific Ocean). The right panels of Fig. B1 shows the differences between the modified latitudinal mean a priori profile and the a priori profiles used for the standard retrieval (a priori from Sect. 4.5). We investigate here retrievals of $H_2O$ and $CH_4$. For $H_2O$ the standard a priori profiles have a large latitudinal dependence, and the difference to the latitudinal mean a priori profile is ocassionally even outside $\pm 200\%$. For $CH_4$ there is also a clear latitudinal dependence in the standard a priori



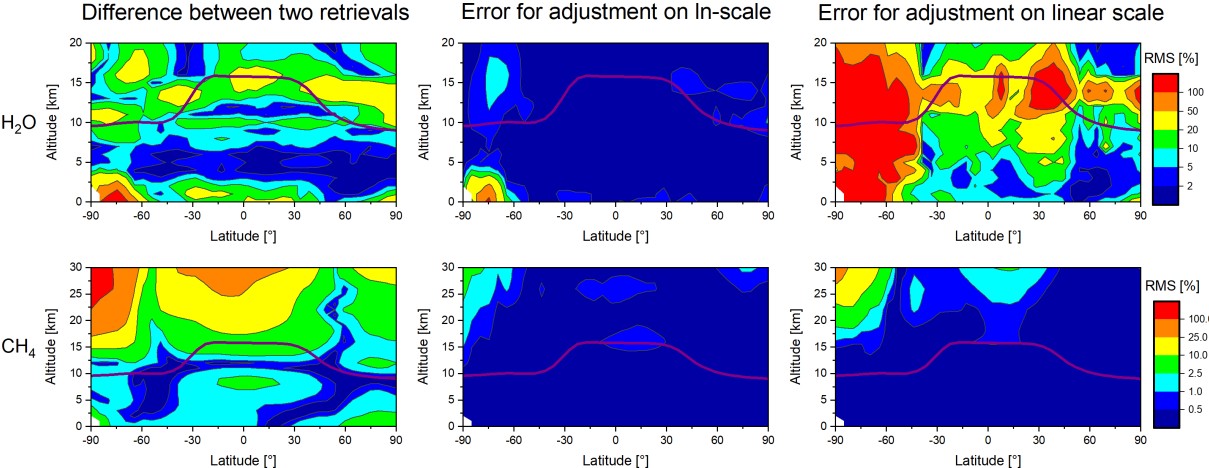

**Figure B2.** Results of the individually performed linearity tests for $H_2O$ and $CH_4$. Shown are the following three latitudinal cross sections: (1) RMS values between the original retrieval (retrieval results using the original a priori data) and the modified retrieval (retrieval results using the modified a priori data), (2) RMS values between the modified retrieval and the original retrieval after performing the a posteriori adjustment according to Eq. (B1) on logarithmic scale, (3) same as (2), but for the a posteriori adjustment performed on linear scale. Thick solid violett line indicates the tropopause altitude.

profiles, which is, however, much smaller than for $H_2O$: below the stratosphere the difference with respect to the latitudinal mean $CH_4$ profile is within $\pm 10\%$.

According to Eq. (A9) we can also simulate the retrieval for the modified a priori by:

$$\hat{x}_m = \hat{x} + (\mathbf{I} - \mathbf{A})(x_{a,m} - x_a). \tag{B1}$$

Here $\hat{x}_m$ is the retrieval results that would be obtained using the modified a priori, $\hat{x}$ is the original retrieval result, $\mathbf{I}$ is the
920 identity matrix, $\mathbf{A}$ the averaging klernel matrix, $x_{a,m}$ the modified a priori, and $x_a$ the original a priori.

The linearity test consists in comparing the results obtained by the full retrieval using the modified a priori data and the results obtained by using the analytic treatment according to Eq. (B1).

### B2    Test results for logarithmic and linear scale

The results of the linearity test are shown in Fig. B2. We demonstrate the impact of the modified a priori by calculating the
925 differences between the original retrieval and the additional retrieval using the modified a priori profiles. We make a latitidinal dependent characterisation of these differences by calculating root-mean-square (RMS) values of the differences within $5°$ latitude bands. Latitudinal cross sections of these RMS differences are depicted on the left of Fig. B2 and reveal that the impact of the modified a priori on the retrieval is largest at the winter polar regions (high southern latitudes). This is where we find large differences between the original and the modified apriori (see Fig. B1) and where at the same time the retrieval sensitivity
is realtively low (see DOFS maps in Fig. 10).





The center and right columns of Fig. B2 show the $5°$ latitude band RMS values for differences between the additional retrieval using the modified a priori profiles and the modification according to the analytic calculations of Eq. (B1). The centre column shows the results when performing the calcuations of Eq. (B1) on the logarithmic scale. We observe that with the analytic calculations we can almost achieve the same results as with the full retrieval calculations. This indicates that the assumption of linearity for such analytic calculation is indeed valid. The right column shows the results when performing the calcuations of Eq. (B1) on the linear scale, i.e. state vectors as well as derivatives (here the averaging kernel entries) are used on the linear scale ($\partial x = x \partial \ln[x]$). The linearity assumption is not valid when performing the analytic calculation for $H_2O$ on the linear scale. We see very large differences between the full retrieval results and the results obtained by Eq. (B1). For $CH_4$ the linear scale analytic calculations also agree poorer to the full retrievals than analytic calculations on logarithmic scale, however, not that pronounced as in case of $H_2O$. The reason is that the $CH_4$ a priori is much weaker modified than the $H_2O$ a priori (see Fig. B1).

In summary, the test shows that the assumption of linearity needed for an analytic treatment of the MUSICA IASI trace gas data is valid. Nevertheless, we have to be careful. Because the retrievals are performed in the logarithmic scale, the analytic calculation that use the averaging kernels, gain matrices, or constraint matrices should also be performed on the logarithmic scale. On this scale the linearity assumption is valid. Contrary to the linear scale, where the linearity assumption is not valid, meaning that an analytic treatment on linear scale can lead to large errors.

## Appendix C: Partial column averaged mixing ratios

For converting mixing ratio profiles into amount profiles we set up a pressure weighting operator $\mathbf{Z}$, as a diagonal matrix with the following entries:

$$Z_{i,i} = \frac{\Delta p_i}{g_i m_{\text{air}}(1 + \frac{m_{H_2O}}{m_{\text{air}}}\hat{x}_i^{H_2O})}. \tag{C1}$$

Using the pressure $p_i$ at atmospheric grid level $i$ we set $\Delta p_1 = \frac{p_2 - p_1}{2} - p_1$, $\Delta p_{nal} = p_{nal} - \frac{p_nal - p_{nal-1}}{2}$, and $\Delta p_i = \frac{p_{i+1} - p_i}{2} - \frac{p_i - p_{i-1}}{2}$ for $1 < i < nal$. Furthermore, $g_i$ is the gravitational acceleration at level $i$, $m_{\text{air}}$ and $m_{H_2O}$ the molecular mass of dry air and water vapour, respectively, and $\hat{x}_i^{H_2O}$ the retrieved water vapour mixing ratio at level $i$.

We define an operator $\mathbf{W}^T$ for resampling fine gridded atmospheric amount profiles into coarse gridded atmospheric partial column amount profiles. It has the dimension $c \times nal$, where $c$ is the number of the resampled coarse atmospheric grid levels and $nal$, the number of atmospheric levels of the original fine atmospheric grid. Each line of the operator has the value '1' for the levels that are resampled and '0' for all other levels:

$$\mathbf{W}^T = \begin{pmatrix} 1 & \cdots & 1 & 0 & \cdots & \cdots & \cdots & \cdots & 0 \\ 0 & \cdots & 0 & 1 & \cdots & 1 & 0 & \cdots & 0 \\ 0 & \cdots & \cdots & \cdots & \cdots & 0 & 1 & \cdots & 1 \end{pmatrix}. \tag{C2}$$

We can combine the operators $\mathbf{Z}$ and $\mathbf{W}^T$ and calculate a pressure weighted resampling operator by:

$$\mathbf{W}^{*T} = (\mathbf{W}^T \mathbf{Z} \mathbf{W})^{-1} \mathbf{W}^T \mathbf{Z}. \tag{C3}$$





This operator resamples linear scale mixing ratio profiles into linear scale partial column averaged mixing ratio profiles.

With operator $\mathbf{W}^{*T}$ we can calculate a coarse gridded partial column averaged state $\hat{\boldsymbol{x}}^*$ from the fine gridded linear mixing ratio state $\hat{\boldsymbol{x}}$ by:

$$\hat{\boldsymbol{x}}^* = \mathbf{W}^{*T}\hat{\boldsymbol{x}}. \tag{C4}$$

Furthermore, we introduce an operator $\mathbf{M}$ for transferring the differentials from logarithmic mixing ratio scale to differentials in linear mixing ratio scale. It is a diagonal matrix having the elements of the linear scale atmospheric mixing ratios state as the diagonal elements:

$$M_{i,i} = \hat{x}_i. \tag{C5}$$

The kernels matrix of the partial column averaged mixing ratio state can then be calculated from the fine gridded logarithmic
scale kernel matrix ($\mathbf{A}$) by

$$\mathbf{A}^* \approx \mathbf{W}^{*T}\mathbf{MAM}^{-1}. \tag{C6}$$

This is here an approximation, because on the right side the diagonal values of $\mathbf{M}$ should be the actual insted of the retrieved mixing ratios.

Similarly the covariances of the partial column averaged mixing ratio state can be calculated from the corresponding covariance matrices of the fine gridded logarithmic scale ($\mathbf{S}$) by

$$\mathbf{S}^* \approx \mathbf{W}^*\mathbf{MSM}^T\mathbf{W}^{*T}. \tag{C7}$$

Here the approximation is because $\Delta x \approx x\Delta\ln x$.

*Author contributions.* Matthias Schneider set up the MUSICA IASI retrieval, designed the netcdf CF conform MUSICA IASI output files, made the calculations in context of the extended output file, developed and performed the compression of the averaging kernel output, and
980 wrote this manuscript. Benjamin Ertl developed the efficient MUSICA IASI processing chain and run the processing at the supercomputer ForHLR. Christopher Diekmann supported the mansucript with several graphics. Farahnaz Khosrawi, Christopher Diekmann, and Benjamin Ertl helped in preparing the MUSICA IASI output files with the compressed averaging kernels. Andreas Weber developed the software tool for compressing the averaging kernels. Frank Hase developed the PROFFIT-nadir retrieval code. Michael Höpfner provided the code use for the MT_CKD water continuum calculations and helped with the scattering calculation needed for the cloud Jacobians. Omaira E. García
985 and Eliezer Sepúlveda provided MUSICA IASI processed data product generated at the Teide supercomputer. Douglas Kinnison provided the CESM1/WACCM data used for generating the MUSICA IASI trace gas a priori data. All authors contributed with corrections and comments to the final version of the manuscript.

*Competing interests.* The authors declare that they have no conflict of interest





*Acknowledgements.* This work has strongly benefit from the project MUSICA (funded by the European Research Council under the European Community's Seventh Framework Programme (FP7/2007-2013)/ERC Grant Agreement number 256961), from financial support in the context of the projects MOTIV and TEDDY (funded by the Deutsche Forschungsgemeinschaft under project IDs/Geschäftszeichen 290612604/GZ:SCHN1126/2-1 and 416767181/GZ:SCHN1126/5-1, respectively), and INMENSE (funded by the Ministerio de Economía y Competividad from Spain, CGL2016-80688-P).

Retrieval calculation for this work were performed on the supercomputer ForHLR funded by the Ministry of Science, Research and the Arts Baden-Württemberg and by the German Federal Ministry of Education and Research. Furthermore, we acknowledge the contribution of Teide High-Performance Computing facilities. TeideHPC facilities are provided by the Instituto Tecnológico y de Energías Renovables (ITER), S.A (teidehpc.iter.es).

We acknowledge the support by the Deutsche Forschungsgemeinschaft and the Open Access Publishing Fund of the Karlsruhe Institute of Technology.




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
