# Peer review of "Design and description of the MUSICA IASI full retrieval product"

_Earth System Science Data, 2021_

## Referee Comment (RC1)

This is a competent, detailed and commendably thorough paper describing the processing of trace gases from IASI, but suffers from being too wordy – there are numerous instances of misspellings and poor sentence structure. In addition to responding to specific comments and corrections listed below, I urge the authors to rewrite the paper to make it more concise.

Corrected spellings shown below in bold type.

Abstract:
There is far too much detail in the abstract. Just make it one or two paragraphs of what's been done. There's no need for something like DOI's and data volumes in the abstract; things like that should stay in the main part of the paper.

Line 47: In **order** to ensure ultimate benefit …

Fig 1: "SSP" is not defined.

Line 91: high spectral **resolution**

Line 92 and elsewhere. Put comma after "e.g." (e.g.**,** Clerbaux et al., 2009).

Line 130: a **tar file** with the **orbit-wise** netcdf files

This number is for the typically 28 orbits per day of **the** two **satellites**.

Lines 135 and 211: Define Jacobians (at least briefly) the first time it's used outside of the abstract.

Line 175: EUMETSAT L2 PPF **version** 6 data are

Line 184: For this **purpose** we use

Line 188: **subtropical** regions, where cloud-free condition**s** generally prevail.

Paragraph at Line 214: "We show trace gas Jacobians for **a** uniform increase of the trace gases throughout the whole atmosphere: 100% for H2O and HDO, 10% for N2O and CH4, 50% for HNO3. The respective values are reasonable approximations to the typical atmospheric variabilities of these trace gases." (note correction)

I was confused by this. A Jacobian is a derivative, not a step change, and it should not be assumed that they are linear much outside the value of the trace gas concentration where they would be calculated. Why wouldn't a 100% change in the H2O be so non-linear that it would give an incorrect Jacobian?

Line 220: Define "a.s.l." the first time it's used.

"Atmospheric temperature variations close to the surface affect mainly the radiances below 1300

cm$^{-1}$ and variations at higher altitudes mainly the radiances above 1300 cm$^{-1}$ ."

Again, this is confusing. If I'm not mistaken, IASI frequencies go as low as 645 cm-1 and can so capture very strong, stratospheric-sensitive lines in the 14 micron CO2 band.

Figure 4: The panels do not show Jacobians, but rather step changes in the radiance from arbitrary changes in a gas' mixing ratio, skin temperature, etc.

Paragraph at line 229: I suggest moving this to where it can be discussed in the context of error calculation.

Line 269: "because they allow  the correct a priori statistics"

Line 299: "Surface skin temperature and the spectral frequency shift are also components of the state vector; however, they are not constrained during the retrieval procedure."

Why not? How would an optimal estimation retrieval work if they are unconstrained?

Line 314: "nudged" is a colloquial term. Suggest using quotes: "the meteorological fields are "nudged" towards meteorological analysis …"

Line 319: **seasonal** cycles

Figure 5 caption: depicted as **violet** thick solid line

Line 334: "Above the troposphere we smoothly connect the tropospheric δD values with the typical stratospheric δD value of −350‰."

This seems too high in the lower stratosphere for δD: Wang et al. (*Remote Sens.* **2018**, *10*, 166; doi:10.3390/rs10020166) report values of -550 to -650‰ depending on latitude.

Figure 7 caption: the a priori assumption **and** the coloured line

Line 440: for all individual **observations**

Line 574: the natural **variability** of δD, N2O,

Line 616: understood by the **forward** model

Line 619: "However, occasionally the measured spectra is very poorly understood by the forward model and the residuals can not be described as a white noise instead the residuals show systematic signatures."

This is unclear. A forward model produces synthetic spectra. How can measured spectra be "understood" by the forward model?

Line 637: The standard files provide for all **observations**

Line 755: we need from the side of the **MUSICA** IASI data,

Line 771: The MUSICA IASI full retrieval product **provides information**

Line 804: on the MUSICA IASI a **priori** settings and constraints

Line 815: Furthermore, the **additional** supply of constraint matrices

Line 817: information available **offers** excellent data reuse

Line 844: "For a general introduction on vector and matrix algebra we recommend dedicated textbooks."

This is annoyingly didactic.

Line 874: the **identity** operator

Figure B1 caption: of the footprints of the used **exemplary** orbit

Figure B1 and B2 captions: thick solid **violet** line

Line 899: used for **modifying** the retrieval settings

Line 914: profile is **occasionally** even outside

Line 920: the averaging **kernel**

Line 930: is **relatively** low

Line 939: Suggest "the linear scale analytic calculations have worse agreement with the full retrievals"

Line 940: "The reason is that the CH4 a priori is much weaker modified than the H2O a priori (see Fig. B1). "

The meaning is unclear.

Line 972: Change "This is here an approximation," to "This is an approximation,"

Suggest changing "should be the actual insted of the retrieved mixing ratios." to "should be the actual mixing ratios instead of those retrieved." ?

---

## Author Comment (AC1)

This is a competent, detailed and commendably thorough paper describing the processing of trace gases from IASI, but suffers from being too wordy – there are numerous instances of misspellings and poor sentence structure. In addition to responding to specific comments and corrections listed below, I urge the authors to rewrite the paper to make it more concise.

We would like to thank the referee for their detailed reading through our long manuscript and apologize for the misspellings and our English (which might not be perfect). We will go through the manuscript and try to shorten where possible without removing important information. In collaboration with the Copernicus Publishing Office we are sure to get a final version with correct English: https://publications.copernicus.org/services/copy_editing_for_english.html

Corrected spellings shown below in bold type.

Abstract:
There is far too much detail in the abstract. Just make it one or two paragraphs of what's been done. There's no need for something like DOI's and data volumes in the abstract; things like that should stay in the main part of the paper.

In the ESSD instructions for the preparation of the manuscript it says:
"**Abstract**: the abstract should be intelligible to the general reader without reference to the text. After a brief introduction of the topic, the summary recapitulates the key points of the article and mentions possible directions for prospective research. Reference citations should not be included in this section (except for data sets) and abbreviations should not be included without explanations. At least for the final accepted publication, a functional data set DOI and its in-text citation must be given in the abstract. If multiple data set DOIs are necessary, please instead refer to the data availability section."

This is why we included the detailed information on the data and DOI in the abstract. However, since we have two DOI's we will remove these details from the abstract (in accordance with the manuscript preparation guidelines and the referee suggestion). Instead, in the abstract we will refer to the data availability section.

Line 47: In **order** to ensure ultimate benefit …
Thanks!

Fig 1: "SSP" is not defined.
S5P is the Sentinel-5 Precursor satellite instrument. We can describe this in the Figure caption, but of course this will make it more wordy.

Line 91: high spectral **resolution**
Thanks!

Line 92 and elsewhere. Put comma after "e.g." (e.g.**,** Clerbaux et al., 2009).
Ok, we changed "e.g." everywhere to "e.g.,", thanks!

Line 130: a **tar file** with the **orbit-wise** netcdf files
This number is for the typically 28 orbits per day of **the** two **satellites**.

We will clarify by changing to: "The typical size of a tar file with the orbit-wise netcdf files of a single day is 15 GB. This number is for the typically 28 orbits per day of two satellite (for three satellites there are typically 42 orbits per day).

We will write in line 135: "[…] detailed information on Jacobians and gain matrices. The Jacobian matrices collect the derivatives of the radiances as measured by the satellite sensors with respect to any parameter (e.g., atmospheric temperature, instrumental conditions) and the gain matrices are the derivatives of the retrieved atmospheric state with respect to the radiances."

Line 175: EUMETSAT L2 PPF **version** 6 data are
Thanks!

Line 184: For this **purpose** we use
Thanks!

Line 188: **subtropical** regions, where cloud-free condition**s** generally prevail.
Thanks!

Paragraph at Line 214: "We show trace gas Jacobians for **a** uniform increase of the trace gases throughout the whole atmosphere: 100% for $H_2O$ and HDO, 10% for $N_2O$ and $CH_4$, 50% for $HNO_3$. The respective values are reasonable approximations to the typical atmospheric variabilities of these trace gases." (note correction)
Thanks!

I was confused by this. A Jacobian is a derivative, not a step change, and it should not be assumed that they are linear much outside the value of the trace gas concentration where they would be calculated. Why wouldn't a 100% change in the $H_2O$ be so non-linear that it would give an incorrect Jacobian?
The referee is right. What we plot and describe is no Jacobian. Actually, we show the Jacobian multiplied by a change in the trace gases of 100%. We will change in the Figure and text "Jacobian" by "Spectral Response".
For the trace gases and for temperature the forward code calculates the derivatives (the Jacobians). For the spectroscopic parameters and for clouds we directly calculate the spectral response by performing two forward calculations with accordingly modified spectroscopic and cloud parameters.

Line 220: Define "a.s.l." the first time it's used.
We will add in parenthesis: "(a.s.l. means above sea level)".

"Atmospheric temperature variations close to the surface affect mainly the radiances below 1300 cm−1 and variations at higher altitudes mainly the radiances above 1300 cm−1 ."
Again, this is confusing. If I'm not mistaken, IASI frequencies go as low as 645 cm-1 and can so capture very strong, stratospheric-sensitive lines in the 14 micron $CO_2$ band.

This is right; however, here we refer to the spectral region analysed by our retrieval. In order to clarify this we will write: "In the analysed 1190 – 1400 cm-1 spectral region, the atmospheric variations […]".

Figure 4: The panels do not show Jacobians, but rather step changes in the radiance from arbitrary changes in a gas' mixing ratio, skin temperature, etc.
This is right. As aforementioned we will change this from Jacobian to "Spectral Response".

Paragraph at line 229: I suggest moving this to where it can be discussed in the context of error calculation.
We don't feel that this fits well to the error discussion. Here we discuss the spectral response, not the error. The error would be the gain matrix multiplied to the spectral response.
Showing and discussing different spectral responses has the purpose to reveal that the interesting spectral signals of trace gas variations are competing with spectral signals caused by not well known parameters (spectroscopic parameters) or cloud coverage.

Line 269: "because they allow considering the correct a priori statistics"
Sorry, we do not understand this comment.

Line 299: "Surface skin temperature and the spectral frequency shift are also components of the state vector; however, they are not constrained during the retrieval procedure."
Why not? How would an optimal estimation retrieval work if they are unconstrained?
Surface skin temperature and spectral frequency shift can be identified very clearly in the spectra. There is no need for imposing a priori information and thereby constraining these retrieved quantities. Also without such constraint the retrieval converges in a very stable manner.

Line 314: "nudged" is a colloquial term. Suggest using quotes: "the meteorological fields are "nudged" towards meteorological analysis …"
Ok, we agree.

Line 319: **seasonal** cycles
Thanks!

Figure 5 caption: depicted as **violet** thick solid line
Thanks!

Line 334: "Above the troposphere we smoothly connect the tropospheric δD values with the typical stratospheric δD value of −350‰."
This seems too high in the lower stratosphere for δD: Wang et al. (*Remote Sens.* **2018**, *10*, 166; doi:10.3390/rs10020166) report values of -550 to -650‰ depending on latitude.
Actually, the used δD a priori profiles are rather similar to Wang et al. (2018). Starting from the surface δD decreases to about -600‰. Above the tropopause the values smoothly increase to about -350‰ in the middle/upper stratosphere (see our Fig. 5). This behavior is rather similar to what is shown in Fig. 2 of Wang et al. (2018). In order to clarify we suggest the following changes (changes in bold): "Above the **tropopause (where δD decays is close to -600‰)** we […]".

Figure 7 caption: the a priori assumption **and** the coloured line
Thanks!

Line 440: for all individual **observations**
Thanks!

Line 574: the natural **variability** of δD, N2O,
Thanks!

Line 616: understood by the **forward** model
Thanks!

Line 619: "However, occasionally the measured spectra is very poorly understood by the forward model and the residuals can not be described as a white noise instead the residuals show systematic signatures."
This is unclear. A forward model produces synthetic spectra. How can measured spectra be "understood" by the forward model?
Thanks, we will replace "understood" by "simulated".

Line 637: The standard files provide for all **observations**
Thanks!

Line 755: we need from the side of the **MUSICA** IASI data,
Thanks!

Line 771: The MUSICA IASI full retrieval product **provides information**
Thanks!

Line 804: on the MUSICA IASI a **priori** settings and constraints
Thanks!

Line 815: Furthermore, the **additional** supply of constraint matrices
Thanks!

Line 817: information available **offers** excellent data reuse
Thanks!

Line 844: "For a general introduction on vector and matrix algebra we recommend dedicated textbooks." This is annoyingly didactic.
Ok, we will remove this sentence.

Line 874: the **identity** operator
Thanks!

Figure B1 caption: of the footprints of the used **exemplary** orbit

Thanks!

Figure B1 and B2 captions: thick solid **violet** line
Thanks!

Line 899: used for **modifying** the retrieval settings
Thanks!

Line 914: profile is **occasionally** even outside
Thanks!

Line 920: the averaging **kernel**
Thanks!

Line 930: is **relatively** low
Thanks!

Line 939: Suggest "the linear scale analytic calculations have worse agreement with the full retrievals"
Thanks, we will change to: "For CH4 the linear scale analytic calculations have worse agreement with the full retrievals than the logarithmic scale calculations; however, […]"

Line 940: "The reason is that the CH4 a priori is much weaker modified than the H2O a priori (see Fig. B1). " The meaning is unclear.
We will clarify this by replacing this sentence by: "The reason is the setup of the linearity test (see Fig. B1 and the corresponding discussion): for the test the modification of the CH4 a priori is weak, but for H2O the a priori modification is rather strong."

Line 972: Change "This is here an approximation," to "This is an approximation," Suggest changing "should be the actual insted of the retrieved mixing ratios." to "should be the actual mixing ratios instead of those retrieved." ?
Ok, we agree with the corrections/suggestions, thanks!

---

## Author Comment (AC2)

We would like to thank the referee for their interest in our work and for revising our manuscript.

The IASI interferometer data are of great significance for global atmospheric research. The authors demonstrated a high professional quality of spectra analysis and state-of-art retrieval algorithm.
Thanks!

A rather large TOTAL data volume is not a serious disadvantage, a real disadvantage is archiving orbit-by-orbit mode. A global coverage is not necessary for many applications, but users have to download global data and delete unnecessary information to save memory. Downloading of ~ 1 TB (one year) may take ~ 50 hours. It may be suggested a simple filtering service of a type as follows.
1) Reading standard .nc file; 2) selecting user-defined list of variables; 3) writing of data in a new .nc file.
We fully agree with the referee that it would be highly desirable to set up a user-friendly data framework that allows the download of user defined MUSICA IASI data packages. Actually, the development and establishment of such framework is exactly the project idea for which we currently try to acquire funds (decision on this proposal is expected by the end of the year).
We hope for the understanding of the referee and the community that this needs additional development work and at this stage we can only declare our firm intention to go into this direction and that we are already intensively working on the acquisition of respective funds.

---

## Author Response (AR1)

Dear Editor,

we have made the corrections in line with our responses to the two referees. Details on the modifications are given in the uploaded track-changes file.

Best regards, Matthias Schneider

---

## Author Response (AR2)

Dear Editor,

in line with your suggestion, we have included the DOIs and references for the data sets in the abstract.

Best regards,

Matthias Schneider